# Target Label-Free Confidence Calibration Under Label Shift

## Abstract

Confidence calibration of classification models is crucial in safety-critical decision-making fields and has received extensive attention. However, general confidence calibration methods rely on the presumption that training and test data are independent and identically distributed ($i.i.d.$), which is often ineffective in real-world data where label shifts often exist. Previous works on confidence calibration under label shift heavily rely on the perception of the target domain label distribution, while the target domain's label distribution is usually unavailable in practice. To overcome this limitation, this paper explores a principled confidence calibration method under label shift that does not require any target domain label information, named Target Label-Free Confidence Calibration (TLFCC), which is realized by utilizing available variables to principledly replace variables related to the label distribution of target domain. Theoretically, this method is proven to achieve approximately correct calibration with high probability, with sample complexity comparable to histogram binning. In addition, this paper proposes a simulation data generation method for confidence calibration under label shift, which can serve as a benchmark to illustrate the discrepancy between the estimated calibration curve and the true calibration curve in the target domain, thereby reflecting the effectiveness of the calibration method. The effectiveness of our calibration method is verified in simulated and real-world data. We believe that our exploration on confidence calibration under label shift will contribute to the development of better-calibrated models, ultimately contributing to the advancement of trustworthy AI.

## 1 Introduction

The accuracy of modern machine learning classification models, like deep neural networks, is consistently on the rise, resulting in their increasing application in various safety-critical fields (Shu et al., 2024; Jiang et al., 2023). Nevertheless, decision-making systems in such fields demand not only high accuracy but also the ability to indicate when they might be incorrect (Munir et al., 2023). For example, in an automatic disease diagnosis system, when the diagnostic model's confidence level is relatively low, it is advisable to delegate the decision-making process to a medical professional (Jiang et al., 2011). Specifically, a classification model should provide accurate confidence alongside its prediction, corresponding to the actual probability of the event occurring. Moreover, accurate confidence offers more detailed information compared to a lack of confidence or merely a class label (Huang et al., 2020). For instance, a statement like "there is a 70% probability that the patient has cancer" provides doctors with more information to make more reliable decisions than just a class label of "cancer". Additionally, accurate confidence enables the integration of classification models into other probabilistic models more effectively. For example, it allows active learning to select more representative samples (Han et al., 2024) and enhances the generalization performance of knowledge distillation (Li & Caragea, 2023). Hence, striving for more accurate confidence in classification models is a task of great significance (Błasiok et al., 2023; Gupta et al., 2021).

Confidence calibration is the most direct solution to obtain more accurate confidence and has been widely studied in recent years (Dong et al., 2025b; Guo et al., 2017; Zhang et al., 2020; Kull et al., 2019). However, general confidence calibration methods rely on the assumption that the target domain (or test set) and the source domain (or calibration set) are independent and identically distributed ($i.i.d.$) (Popordanoska et al., 2024). Real-world settings often violate $i.i.d.$ assumption, i.e.,

the data distribution may shift (Quinonero-Candela et al., 2022). Label shift is a common type of data distribution shift, which refers to the shift occurring in the label distribution of different domains (Garg et al., 2020), often encountered in class-imbalanced data such as epidemic diagnosis (Lipton et al., 2018) and fault detection (Jing et al., 2017). Label shift can bias the posterior probability predicted by the model, leading to inaccurate confidence (Hong et al., 2021). Therefore, it is necessary to study the confidence calibration method under label shift (Sun et al., 2023; Sanchez Aimar et al., 2025).

Previous works that calibrate confidence under label shift heavily rely on obtaining the target label distribution (i.e., label distribution of target domain or test set) and are primarily divided into two groups: 1) Assuming that the target label distribution is known (Hong et al., 2021; Sun et al., 2023); 2) Make additional estimation for the target label distribution (Podkopaev & Ramdas, 2021; Popordanoska et al., 2024). The former provides valuable theoretical exploration for confidence calibration under label shift, but its practicality is limited due to the unavailability of target labels in practice. The latter can be used in practice but brings additional estimation computation and risks of passing the estimation error of target label distribution to confidence calibration, and often rely on the assumption that the confusion matrix is invertible (Lipton et al., 2018; Azizzadenesheli et al., 2019; Garg et al., 2020; Saerens et al., 2002; Ye et al., 2024; Wei et al., 2024).

Therefore, a natural but ignored question is studied: Does confidence calibration under label shift necessarily rely on the target label distribution? Can we develop a calibration method that is independent of the target label distribution under label shift? In fact, when a label shift occurs, the predicted confidence distribution will also change. Can we obtain information from the changed confidence distribution to calibrate the confidence score? Based on this idea, this paper derives a principled calibration method under label shift in the context of predicted-class calibration (Guo et al., 2017; Dong et al., 2025a). This method relies on the available predicted confidence distribution on the target domain rather than the target label distribution. Specifically, this paper first derives the confidence calibration equation under label shift normally and then principledly replaces the unavailable variables related to the target label with other estimable variables. In addition, this paper proposes a new simulated data generation method for confidence calibration under label shift, which can be used as a benchmark to compare the effectiveness of calibration methods by comparing the difference between the estimated calibration curve and the true calibration curve on the simulated data. Our contributions can be summarized as follows:

- We reveal that the label distribution information of the target domain is not necessary for predicted-class confidence calibration under label shift, which provides a new solution idea for confidence calibration under label shift.

- A principled confidence calibration method under label shift that does not require any target domain label information is proposed, named Target Label-Free Confidence Calibration (TLFCC), and is proven to be theoretically and practically feasible.

- A simulation data generation method for confidence calibration under label shift is proposed, which can serve as a benchmark to compare the effectiveness of calibration methods by comparing the discrepancy between the estimated calibration curve and the true calibration curve on the simulated data.

## 2 BACKGROUND AND RELATED WORK

Consider a $K$-class classification problem. The random variable $X \in \mathcal{X}$ represents the input feature and $Y \in \mathcal{Y}$ represents the label variable, where $\mathcal{X} \subset R^d$ and $\mathcal{Y} = \{1, 2, \cdots, K\}$. Let $f \colon \mathcal{X} \to \mathcal{S} \subset \Delta_{K-1}$ be a probabilistic classifier, where $\Delta_{K-1}$ represents a simplex with free-degree $K - 1$. The predicted confidence score vector is $S = f(X) = (S_1, S_2, \cdots, S_K) \in \mathcal{S}$. In general, let $\hat{Y} = \operatorname{argmax}_k \{S_k\}_{1 \le k \le K}$ be predicted class and $\hat{S} = \max \{S_k\}_{1 \le k \le K}$ be the confidence score of predicted class.

We usually focus on the predicted class confidence. This allows a multi-class problem to be reduced to a binary one by defining $H = \mathbf{1}_{Y = \hat{Y}}$, where $\mathbf{1}$ is an indicator: $H = 1$ if $Y = \hat{Y}$, else $H = 0$.

In label shift, let $P$ and $Q$ denote the probability measures of the source domain and the target domain, respectively. $D_s = \{(\hat{s}_i, y_i, \hat{y}_i)\}_{1 \le i \le N_s}^{N_s}$ and $D_t = \{(\hat{s}_i, \hat{y}_i)\}_{1 \le i \le N_t}^{N_t}$ represent the source

domain data and target domain data respectively, where $\hat{s}_i$, $y_i$, and $\hat{y}_i$ represent the observed value of $\hat{S}$, $Y$, and $\hat{Y}$ respectively, and $N_s$ and $N_t$ represent the sample size of the source domain data and target domain respectively. Since the target domain data $D_t$ does not contain the true labels, the considered method is an unsupervised domain adaptation method.

## 2.1 CONFIDENCE CALIBRATION

The purpose of confidence calibration is to match the confidence of the predicted class with the true posterior probability of that class. Formally, we state:

**Definition 1. (Perfect Calibration)** *A classifier is perfectly calibrated if the following equation holds:*

$$Q(Y = \hat{Y}|\hat{S} = \hat{s}) = \hat{s}, \tag{1}$$

*where $\hat{s}$ is the observed value of $\hat{S}$.*

Obviously, Eq. 1 can also be written as $Q(H = 1|\hat{S} = \hat{s}) = \hat{s}$. Typically, $Q(H = 1|\hat{S})$ is called true calibration curve in the target domain.

Recently, confidence calibration has received extensive attention, and existing work can primarily be divided into two groups: train-time calibration (Liu et al., 2023; Müller et al., 2019; Fernando & Tsokos, 2022; Hebbalaguppe et al., 2022; Grathwohl et al., 2020; Yang & Ji, 2021) and post-hoc calibration (Guo et al., 2017, Kull et al., 2019, Zhang et al., 2020, Rahimi et al., 2020, Gupta et al., 2021, Dong et al., 2025b, Zhang & Xie, 2025, Tao et al., 2025). Train-time calibration usually performs calibration during the classifier's training by modifying the objective function, and post-hoc calibration learns a transformation (referred to as a calibration map) of the trained classifier's predictions on a calibration dataset in a post-hoc manner. Although many calibration methods are mentioned above, the effectiveness of these methods relies on the $i.i.d.$ assumption between the target domain and the source domain. When label shift occurs, the $i.i.d.$ assumption breaks down, making it difficult for these methods to achieve effective confidence calibration. Therefore, it is necessary to study the confidence calibration method under label shift.

## 2.2 CONFIDENCE CALIBRATION UNDER LABEL SHIFT

In label shift, the target domain and the source domain have different label distributions but the same class-conditional distributions. Formally, we state:

**Definition 2. (Label Shift)** *Label shift occurs if the following two conditions are satisfied: $P(Y) \neq Q(Y)$ and $P(X|Y) = Q(X|Y)$.*

Label shift will cause the trained model to produce biased posterior probabilities on the target domain, and the details are shown in Appendix A.

In post-hoc calibration, the classifier remains unchanged, i.e., $P(S|X) = Q(S|X)$. By Definition 2, $P(X|Y) = Q(X|Y)$ and then $P(S|Y) = Q(S|Y)$ and $P(\hat{Y}|Y) = Q(\hat{Y}|Y)$. Therefore, the following assumptions hold (Lipton et al., 2018):

**Definition 3. (Post-Hoc Assumption for Label Shift)** *For post-hoc calibration, label shift occurs if the following conditions are satisfied: $P(Y) \neq Q(Y)$, $P(S|Y) = Q(S|Y)$, $P(\hat{Y}|Y) = Q(\hat{Y}|Y)$, and $P(S, \hat{Y}|Y) = Q(S, \hat{Y}|Y)$.*

The purpose of confidence calibration under label shift is to satisfy Definition 1 under label shift. It has received less attention than confidence calibration under the $i.i.d.$ assumption or accuracy improvement under label shift. Hong et al. (2021) proposed a method to decouple the classifier's prediction and the label distribution, named label distribution disentangling (LADE), and empirically proved that LADE can improve the confidence of the target domain. Sun et al. (2023) recalibrated the confidence scores of the classifier to improve the confidence on the target domain. However, both methods rely on the target domain's label distribution that cannot be obtained in practical applications. Podkopaev & Ramdas (2021) and Popordanoska et al. (2024) proposed first estimating the label importance weights using BBSE (Lipton et al., 2018) or RLLS (Azizzadenesheli et al., 2019) and then calibrating the confidence using the estimated label importance weights. However, such methods bring additional estimation computation and risk of passing the estimation error of

target label distribution to confidence calibration, and often rely on the assumption that the confusion matrix is invertible. Therefore, this paper strives to explore a principled confidence calibration method under label shift that does not depend on the target label distribution.

## 3 METHOD

In this section, the following questions are studied: 1) How to perform confidence calibration under label shift? 2) How to eliminate the dependence on label distribution of the target domain during calibration? 3) How about the theoretical properties of the proposed method?

### 3.1 CALIBRATION

**Theorem 1. (Calibration)** *In label shift, the true calibration curve $Q(H = 1|\hat{S})$ on the target domain can be obtained as follows:*

$$Q(H = 1|\hat{S}) = \frac{\sum\limits_{k=1}^{K} P(\hat{S}|Y = k, \hat{Y} = k) \cdot Q(\hat{Y} = k) \cdot Q(H = 1|\hat{Y} = k)}{Q(\hat{S})}. \tag{2}$$

*See Appendix B for proof.*

**Remark on Theorem 1:** The purpose of Theorem 1 is to separate the estimable distributions from difficult-to-estimate distributions (i.e., related to the distribution of target domain labels), so that more effort can be devoted to dealing with difficult-to-estimate distributions later. $P(\hat{S}|Y = k, \hat{Y} = k)$, $Q(\hat{S})$, and $Q(\hat{Y} = k)$ are independent of the target domain labels and are estimable. Specifically, $P(\hat{S}|Y = k, \hat{Y} = k)$ and $Q(\hat{S})$ can be estimated using beta distribution for continuous cases (Dong et al., 2025b; Kull et al., 2017), or using confidence binning for discrete cases (i.e., computing $P(\hat{S} \in b|Y = k, \hat{Y} = k)$ and $Q(\hat{S} \in b)$, where $b$ represents confidence bin). $Q(\hat{Y} = k)$ can be unbiasedly estimated through frequency estimating probability, i.e., $Q(\hat{Y} = k) \approx \hat{N}_t^{(k)}/N_t$, where $\hat{N}_t^{(k)}$ represents sample size predicted by the classifier as $k$-th class and $N_t$ represents the total sample size in the target domain. Therefore, only $Q(H = 1|\hat{Y} = k)$ is unavailable and is related to the label distribution of the target domain. Different from existing confidence calibration methods under label shift that explicitly depend on the target domain's label distribution (Hong et al., 2021; Sun et al., 2023; Podkopaev & Ramdas, 2021; Popordanoska et al., 2024), Theorem 1 implicitly depends on the target domain's label distribution. Therefore, the next task is to replace $Q(H = 1|\hat{Y} = k)$ with other computable probabilities.

### 3.2 TARGET LABEL-FREE CONFIDENCE CALIBRATION

**Theorem 2. (Target Label-Free Calibration)** *In label shift, if $Q(\hat{S}|Y = k, \hat{Y} = k) \neq Q(\hat{S}|Y \neq k, \hat{Y} = k)$, then:*

$$Q(H = 1|\hat{Y} = k) = \frac{Q(\hat{S}|\hat{Y} = k) - P(\hat{S}|Y \neq k, \hat{Y} = k)}{P(\hat{S}|Y = k, \hat{Y} = k) - P(\hat{S}|Y \neq k, \hat{Y} = k)}. \tag{3}$$

*See Appendix C for proof. Therefore, the true calibration curve $Q(H = 1|\hat{S})$ can be obtained by combining Eq. 2:*

$$Q(H = 1|\hat{S}) = \sum_{k=1}^{K} \frac{Q(\hat{Y} = k)P(\hat{S}|Y = k, \hat{Y} = k)}{Q(\hat{S})} \cdot \frac{Q(\hat{S}|\hat{Y} = k) - P(\hat{S}|Y \neq k, \hat{Y} = k)}{P(\hat{S}|Y = k, \hat{Y} = k) - P(\hat{S}|Y \neq k, \hat{Y} = k)}. \tag{4}$$

**Remark on Theorem 2:** Theorem 2 enables the replacement of distributions dependent on target domain labels with those that are estimable. Fundamentally, it exploits the discrepancy in confidence distributions between the source and target domains. All probabilities on the right side of Eq. 4 can be estimated empirically without requiring the target domain label distribution. More importantly,

Eq. 4 does not depend on $P(H = 1|\hat{S})$, i.e., the classifier does not need to be calibrated on the source domain. Perhaps the condition of Theorem 2 may not hold in some special confidence points, i.e. $Q(\hat{S}|Y = k, \hat{Y} = k) = Q(\hat{S}|Y \neq k, \hat{Y} = k)$. However, Theorem 2 can still be used, because most of the confidence points will satisfy the condition of Theorem 2, as shown in Appendix H.3.4. Therefore, we can use interpolation to calibrate calibration for the confidence points that do not satisfy the conditions.

**Empirical Computation:** Eq. 4 shows that the key to estimating $Q(H = 1|\hat{S})$ is to estimate $Q(\hat{Y} = k)$ and the probabilities of confidence $\hat{S}$ under different conditions. The estimation of $Q(\hat{Y} = k)$ can be found in the Remark on Theorem 1. Histogram binning (Freedman & Diaconis, 1981) is a practical and popular method for estimating confidence distributions. Therefore, this paper discusses how to estimate the true calibration curve using histogram binning through Theorem 2. Specifically, $P(\hat{S} \in b_i) \approx \sharp b_i / \sum_{j=1}^{B} \sharp b_j$, where $b_i$ represents the confidence bin and $\sharp b_i$ represents the sample size in $b_i$ on the source domain, and $B$ represents the number of bins. Formally, the final calibration equation is as follows:

$$Q(H = 1|\hat{S} \in b) = \sum_{k=1}^{K} \left[ \frac{Q(\hat{Y} = k)P(\hat{S} \in b|Y = k, \hat{Y} = k)}{Q(\hat{S} \in b)} \cdot Q(H = 1|\hat{Y} = k) \right], \quad (5)$$

where the expression of $Q(H = 1|\hat{Y} = k)$ is as follows:

$$Q(H = 1|\hat{Y} = k) = \frac{Q(\hat{S} \in b|\hat{Y} = k) - P(\hat{S} \in b|Y \neq k, \hat{Y} = k)}{P(\hat{S} \in b|Y = k, \hat{Y} = k) - P(\hat{S} \in b|Y \neq k, \hat{Y} = k)}. \quad (6)$$

In practice, since the classifier is usually well trained, there may be fewer samples with $H = 0$ (i.e., $Y \neq k$ and $\hat{Y} = k$), resulting in a larger estimation error of $P(\hat{S} \in b|Y \neq k, \hat{Y} = k)$. In this case, we recommend using the following formula to calculate $P(\hat{S} \in b|Y \neq k, \hat{Y} = k)$.

$$P(\hat{S} \in b|Y \neq k, \hat{Y} = k) = \frac{P(Y \neq k, \hat{Y} = k|\hat{S} \in b)P(\hat{S} \in b)}{P(Y \neq k, \hat{Y} = k)}$$
$$= \frac{\left( P(\hat{Y} = k|\hat{S} \in b) - P(Y = k, \hat{Y} = k|\hat{S} \in b) \right) P(\hat{S} \in b)}{P(\hat{Y} = k) - P(Y = k, \hat{Y} = k)}. \quad (7)$$

Because Eq. 7's numerator must be 0 when Eq. 7's denominator is 0, there is only a need to add a minimum positive constant to the denominator to prevent it from being 0, rather than exclude confidence score points whose denominator is 0. The pseudo code of Theorem 2's empirical computation process is shown in Algorithm 1 in Appendix G.

### 3.3 THEORETICAL GUARANTEE

Theorem 3 gives the theoretical guarantee of Theorem 2's empirical computation, and its proofs are given in Appendix D. It tells us that the calibration error will be small enough when the sample size is sufficient. In addition, its sample efficiency is similar to that of popular histogram binning (Kumar et al., 2019), both being $\mathcal{O}\left(\frac{B}{\varepsilon^2}\ln\left(\frac{2B}{\delta}\right)\right)$. Therefore, it has broad application potential like histogram binning.

**Theorem 3.** *Let* $D_s^{(k)} = \{(\hat{s}, y, \hat{y})|y = k, \hat{y} = k, (\hat{s}, y, \hat{y}) \in D_s\}$ *and* $\hat{D}_t^{(k)} = \{(\hat{s}, \hat{y})|\hat{y} = k, (\hat{s}, \hat{y}) \in D_t\}$. $\forall \varepsilon > 0$ *and* $\delta \in (0, 1)$, *when* $\min\{\sharp D_s^{(k)}, \sharp \hat{D}_t^{(k)}\} = \mathcal{O}\left(\frac{B}{\varepsilon^2}\ln\left(\frac{2B}{\delta}\right)\right)$, *it holds with probability* $1 - \delta$:

$$\left| \hat{Q}(H = 1|\hat{S} \in b) - Q(H = 1|\hat{S} \in b) \right| \leq \varepsilon, \quad (8)$$

*where* $\hat{Q}(H = 1|\hat{S} \in b)$ *is the calibration result estimated by Theorem 2's empirical computation. See Appendix D for proof.*

## 4 SIMULATING DATASETS

A key challenge in developing confidence calibration is the lack of ground truth for the calibration curve, hindering the comparison of the true calibration curve with the estimated calibration curve.

For the confidence calibration task in $i.i.d.$ data, Roelofs et al. (2022) and Dong et al. (2025b) proposed methods to compare the true calibration curve with the estimated calibration curve. Roelofs et al. (2022) use the fitted function on the publicly available logit datasets as the true calibration curve behind the data. Dong et al. (2025b) preset a calibration curve to generate a simulation dataset by binomial process modeling, and then compare the estimated calibration curve on the generated dataset with the preset true calibration curve. Both methods provide valuable experience for evaluating calibration effect but are not suitable for calibration under label shift because true calibration curves between the target and source domains are not equal in this case.

Therefore, this section proposes a new method for generating simulated data for confidence calibration under label shift. Firstly, we theoretically derived the relationship between the true calibration curves in the source and target domains, and the relationship is used to preset realistic true calibration curves in both domains. Secondly, integrated with the simulated data generation method proposed by Dong et al. (2025b), realistic simulated data under label shift are generated, and then calibration methods are performed on the simulated data to obtain the estimated calibration curves. Finally, the calibration effectiveness can be known by comparing the true and the estimated calibration curves.

**Preset True Calibration Curves:** Taking binary classification as an example. Due to Definition 3, $\hat{S}|Y = 0$ and $\hat{S}|Y = 1$ are invariant between the target domain and the source domain. In addition, due to Theorem 4, $P(H = 1|\hat{S}, Y = k) = Q(H = 1|\hat{S}, Y = k)$. Therefore, Theorem 5 shows the difference between true calibration curves on the source domain and the target domain. Obviously, the difference between Eq. 9 and Eq. 10 is that the label distribution is different, which is consistent with the definition of label shift. When simulation, $\hat{S}|Y = 0$ and $\hat{S}|Y = 1$ can be preset to the beta distribution (Roelofs et al., 2022; Dong et al., 2025b), and $P(H = 1|\hat{S}, Y = k)$ can be preset to the calibration curve functions provided by Dong et al. (2025b). The pseudo code of simulation data generation method is shown in Algorithm 2 in Appendix G. Fig. 1 shows the difference in the preset true calibration curves between the source and target domains. It demonstrates that label shift will bring obvious changes to the true calibration curve.

**Theorem 4.** *In label shift, $P(H = 1|\hat{S}, Y = k) = Q(H = 1|\hat{S}, Y = k)$, where $k \in \mathcal{Y}$. See Appendix E for proof.*

**Theorem 5.** *In label shift, it holds that:*

$$P(H = 1|\hat{S}) = P(H = 1|\hat{S}, Y = 0)\frac{P(\hat{S}|Y = 0)P(Y = 0)}{P(\hat{S}|Y = 0)P(Y = 0) + P(\hat{S}|Y = 1)P(Y = 1)}$$
$$+P(H = 1|\hat{S}, Y = 1)\frac{P(\hat{S}|Y = 1)P(Y = 1)}{P(\hat{S}|Y = 0)P(Y = 0) + P(\hat{S}|Y = 1)P(Y = 1)}, \tag{9}$$

*and:*

$$Q(H = 1|\hat{S}) = P(H = 1|\hat{S}, Y = 0)\frac{P(\hat{S}|Y = 0)Q(Y = 0)}{P(\hat{S}|Y = 0)Q(Y = 0) + P(\hat{S}|Y = 1)Q(Y = 1)}$$
$$+P(H = 1|\hat{S}, Y = 1)\frac{P(\hat{S}|Y = 1)Q(Y = 1)}{P(\hat{S}|Y = 0)Q(Y = 0) + P(\hat{S}|Y = 1)Q(Y = 1)}. \tag{10}$$

*See Appendix F for proof.*

## 5 RESULTS

The effectiveness of the proposed method is verified from three complementary perspectives: 1) On simulated label shift datasets (see section 4), we compare the estimated calibration curve against the true calibration curve in the target domain; 2) We compare calibration metrics of multiple methods on real-world label shift datasets; 3) Through ablation studies, we analyze the impact of key components and design choices to further understand the robustness and performance of our approach.

### 5.1 CALIBRATION ON SIMULATED DATASETS

**Experimental Setup:** To gain a true insight into the calibration effectiveness of the proposed method, we generate realistic simulated data using Algorithm 2 and compare true calibration curves

and estimated calibration curves on the simulated data. Referring to the fitting results of Roelofs et al. (2022) and the preset schemes of Dong et al. (2025b), we select six preset schemes (named D1, D2,···, and D6, respectively), as shown in Table 4 of Appendix H.1.2. The label distribution is set as follows: $P(Y = 0)$ is randomly selected from [0.7, 0.8, 0.9], and $Q(Y = 0)$ is randomly selected from [0.2, 0.3, 0.4].

**Experimental Results:** Fig. 1 shows the results on the simulated data. Firstly, by comparing the true calibration curves on the target and source domain, it shows that label shift leads to a significant change in the true calibration curve, illustrating the necessity of confidence calibration under label shift. Secondly, the calibration curve estimated by our method is closer to the true calibration curve of the target domain than the true calibration curve of the source domain, which verifies that our method can indeed calibrate the confidence of the target domain. Thirdly, the estimated calibration curve fluctuates around the true calibration curve of the target domain with slight errors, which are caused by the density estimation error in Algorithm 1 of Appendix G. Therefore, it will further benefit from more accurate density estimation methods in the future. Finally, when the confidence score is low, e.g., when the confidence score is below 0.4 in D1, D4, and D5, the error of the estimated calibration curve will be larger, which is caused by the larger empirical error due to the sparse samples in low confidence score regions. This problem is common in most calibration methods, but since few samples fall in low-confidence regions, its impact is minimal (Dong et al., 2025b).

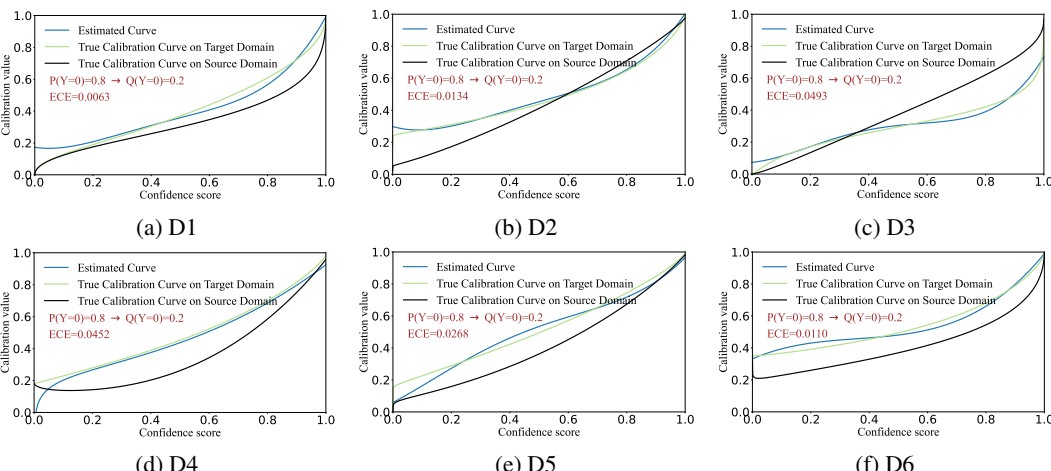

Figure 1: Calibration effectiveness on simulated datasets. Code: https://github.com/ Anonymous-user-code/TLFCC/blob/main/Cali_in_simulated_data.ipynb.

## 5.2 CALIBRATION ON REAL-WORLD DATASETS

### 5.2.1 EXPERIMENTAL SETUP

**Datasets and Networks**: To reflect the effectiveness of calibration methods on the real-world dataset, seven datasets of different types and sizes are selected for experiments: a popular binary tabular recognition dataset **German Credit** (Hofmann, 1994), a seven-class tabular recognition dataset **Dry Bean** (Koklu & Ozkan, 2020), a binary medical image recognition dataset named **MHIST** (Wei et al., 2021), a ten-class digit recognition dataset named **SVHN** (Netzer et al., 2011), a ten-class image recognition dataset named **CIFAR-10** (Krizhevsky et al., 2009), a 100-class image recognition dataset named **CIFAR-100** (Krizhevsky et al., 2009), and a large-scale color real-world image recognition dataset **ImageNet-1K** (Deng et al., 2009). Details of how these datasets are sampled into label shift datasets are given in the Appendix H.1.3, and details of how to select a neural network classifier for these data are given in the Appendix H.1.4.

**Baseline Methods**: To more comprehensively assess the effectiveness of the proposed method, the following methods are compared: 1) **Uncal**: uncalibrated model trained on source data; 2) **TempScal**: calibration on source data using Temperature scaling (Guo et al., 2017); 3) **PCS**: train-time calibration with post-compensated softmax (Hong et al., 2021), where target label distribution

is estimated by RLLS (Azizzadenesheli et al., 2019); 4) **LADE**: a train-time calibration method (Hong et al., 2021), where target label distribution is estimated by RLLS (Azizzadenesheli et al., 2019); 5) **MRR**: post-hoc calibration using Eq. 11 (Sun et al., 2023), where target label distribution is estimated by RLLS (Azizzadenesheli et al., 2019); 6) **LaSCal**: a post-hoc calibration method under label shift (Popordanoska et al., 2024); 7) **TLFCC**: the proposed method.

**Calibration Metrics**: Since TLFCC is a post-hoc calibration method that does not modify the classifier, its classification accuracy remains unchanged. Therefore, we focus on calibration metrics. Three popular calibration metrics are used to compare calibration methods: expected calibration error($ECE_{bin}$) (Guo et al., 2017), debiased calibration error ($ECE_{debiased}$) (Kumar et al., 2019), and calibration error using Kolmogorov-Smirnov test ($KS\text{-}error$) (Gupta et al., 2021).

### 5.2.2 Experimental Results

Table 1: Compare calibration errors on real-world data. "Res" is ResNet (He et al., 2016), "W-Res" is Wide-ResNet (Zagoruyko & Komodakis, 2016), "Dense" is DenseNet (Huang et al., 2017), and "ViT-L" is ViT-Large (Dosovitskiy et al., 2021). "$0.8 \rightarrow 0.4$" indicates $P(Y = 0) = 0.8$ and $Q(Y = 0) = 0.4$. The reported results are mean ± std over ten runs.

| Dataset | $\mathbf{ECE_{bin}}(\%)\downarrow$ | | | | | | |
| --- | --- | --- | --- | --- | --- | --- | --- |
| | Uncal | TempScal | PCS | LADE | MRR | LaSCal | TLFCC |
| ***German Credit*** | | | | | | | |
| LeNet-1D | $36.70_{\pm0.93}$ | $14.00_{\pm0.41}$ | $10.13_{\pm0.43}$ | $13.76_{\pm0.40}$ | $10.47_{\pm0.40}$ | $9.276_{\pm0.44}$ | $7.886_{\pm0.31}$ |
| MLP | $29.64_{\pm1.27}$ | $14.84_{\pm0.53}$ | $14.75_{\pm0.46}$ | $13.04_{\pm0.63}$ | $14.71_{\pm0.72}$ | $11.25_{\pm0.47}$ | $10.42_{\pm0.29}$ |
| TabNet | $36.70_{\pm0.96}$ | $14.00_{\pm0.68}$ | $13.80_{\pm0.44}$ | $9.963_{\pm0.46}$ | $10.47_{\pm0.41}$ | $9.276_{\pm0.28}$ | $7.886_{\pm0.31}$ |
| ***Dry Bean*** | | | | | | | |
| LeNet-1D | $64.33_{\pm2.97}$ | $42.93_{\pm1.34}$ | $40.61_{\pm1.02}$ | $32.86_{\pm1.40}$ | $7.715_{\pm0.24}$ | $0.927_{\pm0.03}$ | $0.650_{\pm0.02}$ |
| MLP | $63.82_{\pm2.65}$ | $41.88_{\pm1.88}$ | $29.73_{\pm1.08}$ | $27.41_{\pm1.35}$ | $8.881_{\pm0.29}$ | $0.956_{\pm0.03}$ | $0.945_{\pm0.04}$ |
| TabNet | $64.88_{\pm1.88}$ | $50.45_{\pm1.28}$ | $26.34_{\pm0.72}$ | $46.03_{\pm1.87}$ | $6.968_{\pm0.19}$ | $1.344_{\pm0.03}$ | $0.944_{\pm0.02}$ |
| ***MHIST-LS*** | | | | | | | |
| Res18 ($0.8 \rightarrow 0.4$) | $23.11_{\pm0.66}$ | $11.31_{\pm0.55}$ | $6.566_{\pm0.24}$ | $9.920_{\pm0.28}$ | $6.650_{\pm0.18}$ | $5.025_{\pm0.22}$ | $4.505_{\pm0.15}$ |
| Res50 ($0.7 \rightarrow 0.4$) | $24.40_{\pm1.19}$ | $8.126_{\pm0.28}$ | $4.258_{\pm0.12}$ | $3.745_{\pm0.12}$ | $3.226_{\pm0.14}$ | $2.548_{\pm0.08}$ | $2.452_{\pm0.10}$ |
| Res101 ($0.9 \rightarrow 0.3$) | $26.10_{\pm1.02}$ | $3.509_{\pm0.13}$ | $2.889_{\pm0.10}$ | $2.365_{\pm0.07}$ | $1.128_{\pm0.04}$ | $1.050_{\pm0.04}$ | $0.390_{\pm0.01}$ |
| ***SVHN-LS*** | | | | | | | |
| Res20 (IF = 2) | $48.20_{\pm2.10}$ | $21.35_{\pm0.82}$ | $15.42_{\pm0.55}$ | $19.87_{\pm0.73}$ | $8.640_{\pm0.31}$ | $2.210_{\pm0.08}$ | $1.320_{\pm0.05}$ |
| Res56 (IF = 5) | $44.10_{\pm1.95}$ | $18.92_{\pm0.77}$ | $13.08_{\pm0.49}$ | $16.74_{\pm0.66}$ | $7.510_{\pm0.28}$ | $1.980_{\pm0.07}$ | $1.170_{\pm0.04}$ |
| Res110 (IF = 10) | $41.75_{\pm1.88}$ | $17.10_{\pm0.73}$ | $12.11_{\pm0.46}$ | $15.02_{\pm0.61}$ | $6.980_{\pm0.26}$ | $1.860_{\pm0.06}$ | $1.050_{\pm0.04}$ |
| ***CIFAR-10-LS*** | | | | | | | |
| Res20 (IF = 2) | $72.59_{\pm3.04}$ | $36.52_{\pm0.98}$ | $25.03_{\pm1.04}$ | $30.76_{\pm1.38}$ | $6.971_{\pm0.18}$ | $0.897_{\pm0.03}$ | $0.570_{\pm0.02}$ |
| Res56 (IF = 5) | $65.59_{\pm1.85}$ | $43.65_{\pm2.10}$ | $5.692_{\pm0.23}$ | $7.640_{\pm0.23}$ | $10.83_{\pm0.38}$ | $3.545_{\pm0.12}$ | $0.785_{\pm0.03}$ |
| Res110 (IF = 10) | $71.54_{\pm2.75}$ | $27.82_{\pm0.93}$ | $7.066_{\pm0.19}$ | $18.18_{\pm0.81}$ | $9.162_{\pm0.43}$ | $2.513_{\pm0.10}$ | $1.110_{\pm0.05}$ |
| ***CIFAR-100-LS*** | | | | | | | |
| Res20 (IF = 2) | $89.40_{\pm3.65}$ | $52.65_{\pm2.40}$ | $38.72_{\pm1.58}$ | $57.31_{\pm2.25}$ | $27.40_{\pm1.28}$ | $9.860_{\pm0.42}$ | $8.230_{\pm0.35}$ |
| Res56 (IF = 5) | $84.10_{\pm3.40}$ | $50.18_{\pm2.28}$ | $34.11_{\pm1.46}$ | $53.04_{\pm2.08}$ | $25.55_{\pm1.19}$ | $9.210_{\pm0.40}$ | $7.940_{\pm0.32}$ |
| Res110 (IF = 10) | $86.75_{\pm3.48}$ | $51.07_{\pm2.33}$ | $35.63_{\pm1.49}$ | $54.22_{\pm2.12}$ | $26.10_{\pm1.22}$ | $9.430_{\pm0.41}$ | $8.010_{\pm0.33}$ |
| ***ImageNet-LS*** | | | | | | | |
| W-Res50 (IF=2) | $59.38_{\pm1.56}$ | $39.33_{\pm1.77}$ | $35.71_{\pm1.43}$ | $31.07_{\pm0.81}$ | $39.17_{\pm1.93}$ | $23.31_{\pm1.05}$ | $7.900_{\pm0.27}$ |
| Dense162 (IF=5) | $82.68_{\pm2.37}$ | $55.53_{\pm1.48}$ | $14.70_{\pm0.43}$ | $51.55_{\pm2.14}$ | $23.09_{\pm1.11}$ | $7.944_{\pm0.34}$ | $7.689_{\pm0.34}$ |
| ViT-L (IF=10) | $78.40_{\pm2.98}$ | $63.26_{\pm3.13}$ | $33.71_{\pm1.07}$ | $36.78_{\pm1.04}$ | $28.96_{\pm0.75}$ | $10.72_{\pm0.38}$ | $7.090_{\pm0.29}$ |

Table 1 reports the computed results of expected calibration error between our calibration method and other calibration methods on the public datasets. See Appendix H.2 for the results in $ECE_{debiased}$ and $KS\text{-}error$. All considered calibration methods perform better than the uncalibrated model, indicating they all can significantly improve confidence. Table 1 demonstrates that TLFCC consistently achieves the lowest calibration error across all real-world datasets and network

architectures, outperforming source-domain calibration methods (e.g., TempScal) and label-shift-aware baselines (PCS, LADE, MRR, LaSCal). The improvement is particularly pronounced under relatively severe label shift, such as CIFAR-10-LS with an imbalance factor of 10, where TLFCC reduces $ECE_{bin}$ by over 50% compared to LaSCal. On large-scale ImageNet-LS, TLFCC also delivers substantial gains, lowering error from 23.31% to 7.90% on Wide-ResNet and from 10.72% to 7.09% on ViT-Large. These results confirm that TLFCC provides robust and scalable calibration without requiring any target-domain label information.

### 5.3 ABLATION EXPERIMENTS

**Impact of Shift Magnitude:** Table 2 shows that calibration error increases as the label shift becomes more severe, where the dataset is MNIST-LS and more results are shown in Appendix H.3.2. When the shift magnitude grows from "$0.6 \to 0.4$" to "$0.99 \to 0.01$", $ECE_{bin}$ rises across all methods. Among the baselines, LaSCal generally performs better than LADE and MRR, but its error still grows notably under extreme imbalance. TLFCC consistently achieves the lowest ECE under all shift magnitudes, which confirms that TLFCC remains robust even when label shift is extreme.

Table 2: Impact of Shift Magnitude. The first column represents: $P(Y = 0) \to Q(Y = 0)$. The classifier is ResNet18.

| Magnitude | ECE$_{\mathbf{bin}}$ (%) $\downarrow$ | | | |
| --- | --- | --- | --- | --- |
| | LADE | MRR | LaSCal | TLFCC |
| $0.6 \to 0.4$ | $9.54_{0.36}$ | $6.05_{0.29}$ | $5.25_{0.25}$ | $4.07_{0.19}$ |
| $0.7 \to 0.3$ | $10.3_{0.38}$ | $6.86_{0.36}$ | $5.43_{0.33}$ | $4.96_{0.26}$ |
| $0.8 \to 0.2$ | $11.3_{0.43}$ | $6.49_{0.43}$ | $6.21_{0.32}$ | $5.19_{0.22}$ |
| $0.9 \to 0.1$ | $12.8_{0.61}$ | $7.41_{0.30}$ | $6.32_{0.25}$ | $5.42_{0.25}$ |
| $0.95 \to 0.05$ | $13.1_{0.75}$ | $7.67_{0.50}$ | $6.90_{0.37}$ | $6.53_{0.29}$ |
| $0.99 \to 0.01$ | $15.2_{0.67}$ | $8.10_{0.49}$ | $7.24_{0.36}$ | $7.12_{0.36}$ |

**Impact of Estimation Methods:** Table 3 compares different estimation strategies for confidence distribution (Line 8 in Algorithm 1) and calibration curve fitting (Line 28 in Algorithm 1), using MNIST-LS with ResNet18 (see Appendix H.3.1 for more results). All combinations yield similar performance, with $ECE_{bin}$ between 4.505% and 4.524%, showing TLFCC's robustness to estimation choices. The default (HB + GLM) performs slightly best, but alternatives cause minimal degradation.

Table 3: Impact of Estimation Methods. **BETA**: beta distribution estimation; **HB**: histogram binning; **CSS**: cubic smoothing spline: **GLM**: generalized linear fitting.

| BETA | HB | CSS | GLM | ECE$_{\mathbf{bin}}$ (%) $\downarrow$ |
| --- | --- | --- | --- | --- |
| ✓ | | ✓ | | $4.524_{0.19}$ |
| ✓ | | | ✓ | $4.521_{0.18}$ |
| | ✓ | ✓ | | $4.516_{0.20}$ |
| | ✓ | | ✓ | $4.505_{0.13}$ |

**Impact of Sample Size:** Fig. 2 shows the effect of target domain sample size on TLFCC. More results are shown in Appendix H.3.3. In addition, Appendix H.3.3 also shows the impact of the source domain sample size. As sample size grows, estimated calibration curve approaches the true target-domain curve. Even with 500 samples, it is relatively close to the true curve.

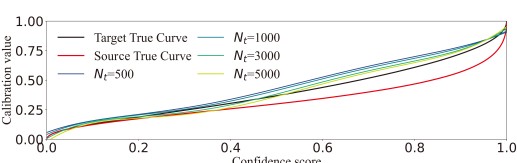

Figure 2: Impact of Target Sample Size on D1.

## 6 DISCUSSION

**Plausibility of Label Shift Assumption:** Label shift is common rather than restrictive: in healthcare, disease prevalence changes across seasons or outbreaks while class-conditional symptom patterns remain stable (Guo et al., 2020); in risk monitoring, fraud/defect rates drift with policy and market cycles even when the within-class signatures are similar (Zhang et al., 2025). Crucially, the same methodology for label shift directly benefits the widely studied class imbalance/long-tailed setting, which is also widely found in real-world problems such as rare disease diagnosis and fault detection (Dong et al., 2025a; Zhang et al., 2023). Therefore, label shift is not just a theoretically valuable assumption, but also a realistic, high-utility assumption that enables measurable benefits in exactly the scenarios practitioners face.

**Underlying Reasons for Method Effectiveness:** The underlying reason why our method can break free from dependence on the target domain label distribution is that the change information in the

confidence distribution can compensate for the lack of label distribution information, as shown in Theorem 2. Compared to the existing state-of-the-art method LaSCal, the underlying reason for our method's success may lie in the fact that LaSCal requires post-training with temperature scaling, which introduces learning errors (such as the impact of temperature scaling's limited expressive power) in addition to estimation errors. In contrast, our method is a principled method that does not require post-training, and the error originates solely from estimation errors.

**Potential Impact, Limitations, and Future Work:** We explore the possibility of not utilizing target domain label information for confidence calibration under label shift. We also propose a solution as a starting point for performing unsupervised domain adaptation calibration under label shift. We believe this method has the potential to inspire a wealth of follow-up research, ultimately enhancing decision-making in real-world applications—particularly for underrepresented populations and safety-critical scenarios. However, our study also has several limitations. Our method may lead to significant errors in areas of low confidence due to the sparsity of samples. Although other existing methods also have this problem, exploring more accurate confidence in sample-sparse areas remains an attractive future research direction. In addition, we did not address covariate shift. Although the pure label shift is common and has been widely studied in prior literature (Hong et al., 2021; Sun et al., 2023), handling mixed shifts (label + covariate) remains an important future direction.

# 7 CONCLUSION

This paper addresses confidence calibration under label shift without target-domain labels. We derive a principled confidence calibration method that only requires estimating the predicted confidence distribution on the target domain, without leveraging any label information from the target domain. Theoretically, it achieves near-correct calibration with high probability and sample complexity comparable to histogram binning. We also introduce a simulation-based benchmark for evaluating calibration methods by comparing estimated and true calibration curves. Extensive experiments on real and simulated data validate our method and demonstrate its effectiveness.

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

# APPENDIX

## A  LABEL SHIFT BACKGROUND

A real-world example of label shift is that the incidence rate of the epidemic changes, but the symptoms do not change during an epidemic outbreak (Guo et al., 2020). By simple Bayes rule derivation, Eq. 11 tells us that label shift will cause the trained model to produce biased posterior probabilities on the target domain (Šipka et al., 2022, Liang et al., 2025, Alaiz-Rodríguez et al., 2009, Moreno-Torres et al., 2012, Hong et al., 2021, Sun et al., 2023, Podkopaev & Ramdas, 2021), resulting in inaccurate prediction and confidence:

$$Q(Y = k|X) = \frac{P(Y = k|X)\frac{Q(Y=k)}{P(Y=k)}}{\sum\limits_{k'=1}^{K} P(Y = k'|X)\frac{Q(Y=k')}{P(Y=k')}}. \tag{11}$$

If we have prior knowledge about $Q(Y)$ or $Q(Y)/P(Y)$, we can adjust the trained model for the target domain from Eq. 11. Therefore, existing methods primarily estimate $Q(Y)$ or $Q(Y)/P(Y)$ from three aspects: 1) Distribution matching method (e.g., feature matching (Zhang et al., 2013; Guo et al., 2020), moment matching (Tian et al., 2023), class probability matching (Wen et al., 2024)); 2) Maximize the likelihood function of the feature distribution in target domain (Saerens et al., 2002; Alexandari et al., 2020); 3) Invert confusion matrix (Lipton et al., 2018; Azizzadenesheli et al., 2019). However, these works aim to improve the classifier's predictive accuracy on the label-shifted domain without addressing its calibration (Popordanoska et al., 2024). Therefore, it is necessary to further study confidence calibration methods under label shift to achieve credible predictions in the target domain.

## B  PROOF OF THEOREM 1

*Proof.* Our goal is to obtain the true calibration curve $Q(H = 1|\hat{S})$ on the target domain. By Bayes' theorem:

$$Q(H = 1|\hat{S}) = \frac{Q(\hat{S}|H = 1) \cdot Q(H = 1)}{Q(\hat{S})}. \tag{12}$$

Then, according to the total probability theorem:

$$\begin{aligned}
Q(\hat{S}|H = 1) &= Q(\hat{S}|Y = \hat{Y}) = \sum_{k=1}^{K} Q(\hat{S}, \hat{Y} = k|Y = \hat{Y}) \\
&= \sum_{k=1}^{K} Q(\hat{S}|Y = \hat{Y}, \hat{Y} = k)Q(\hat{Y} = k|Y = \hat{Y}) \\
&= \sum_{k=1}^{K} Q(\hat{S}|Y = k, \hat{Y} = k)Q(\hat{Y} = k|H = 1).
\end{aligned} \tag{13}$$

Due to Definition 3, the following holds:

$$\begin{aligned}
Q(\hat{S}|Y = k, \hat{Y} = k) &= \frac{Q(\hat{S}, \hat{Y} = k|Y = k)}{Q(\hat{Y} = k|Y = k)} = \frac{P(\hat{S}, \hat{Y} = k|Y = k)}{P(\hat{Y} = k|Y = k)} \\
&= P(\hat{S}|Y = k, \hat{Y} = k).
\end{aligned} \tag{14}$$

Therefore:

$$Q(\hat{S}|H=1) = \sum_{k=1}^{K} P(\hat{S}|Y=k,\hat{Y}=k)Q(\hat{Y}=k|H=1). \tag{15}$$

Substituting Eq. 15 into Eq. 12:

$$
\begin{aligned}
Q(H=1|\hat{S}) &= \frac{\sum\limits_{k=1}^{K} P(\hat{S}|Y=k,\hat{Y}=k) \cdot Q(\hat{Y}=k|H=1) \cdot Q(H=1)}{Q(\hat{S})} \\
&= \frac{\sum\limits_{k=1}^{K} P(\hat{S}|Y=k,\hat{Y}=k) \cdot Q(\hat{Y}=k) \cdot Q(H=1|\hat{Y}=k)}{Q(\hat{S})}.
\end{aligned}
\tag{16}
$$

$\square$

## C  PROOF OF THEOREM 2

*Proof.* According to the total probability formula:

$$
\begin{aligned}
Q(\hat{S}|\hat{Y}=k) &= Q(\hat{S}, H=1|\hat{Y}=k) + Q(\hat{S}, H=0|\hat{Y}=k) \\
&= Q(\hat{S}|H=1,\hat{Y}=k)Q(H=1|\hat{Y}=k) + Q(\hat{S}|H=0,\hat{Y}=k)Q(H=0|\hat{Y}=k) \\
&= Q(\hat{S}|Y=k,\hat{Y}=k)Q(H=1|\hat{Y}=k) + Q(\hat{S}|Y\neq k,\hat{Y}=k)(1 - Q(H=1|\hat{Y}=k)).
\end{aligned}
\tag{17}
$$

Because $Q(\hat{S}|Y=k,\hat{Y}=k) \neq Q(\hat{S}|Y\neq k,\hat{Y}=k)$, $Q(\hat{S}|Y=k,\hat{Y}=k) - Q(\hat{S}|Y\neq k,\hat{Y}=k) \neq 0$. Solve Eq. 17 to obtain:

$$
\begin{aligned}
Q(H=1|\hat{Y}=k) &= \frac{Q(\hat{S}|\hat{Y}=k) - Q(\hat{S}|Y\neq k,\hat{Y}=k)}{Q(\hat{S}|Y=k,\hat{Y}=k) - Q(\hat{S}|Y\neq k,\hat{Y}=k)} \\
&= \frac{Q(\hat{S}|\hat{Y}=k) - P(\hat{S}|Y\neq k,\hat{Y}=k)}{P(\hat{S}|Y=k,\hat{Y}=k) - P(\hat{S}|Y\neq k,\hat{Y}=k)},
\end{aligned}
\tag{18}
$$

where the second equality is due to Eq. 14. $\square$

## D  PROOF OF THEOREM 3

*Proof.* Let $\sharp b_i$ represent the sample size of bin $b_i$, $B$ represent the number of bins, and $F_{b_i} = \sharp b_i / \sum_{j=1}^{B} \sharp b_j$. By Hoeffding's inequality, it holds that $\forall \varepsilon > 0$ and sample size $N_s$ in domain source, then:

$$P\left(\left|F_{b_i} - P(\hat{S} \in b_i)\right| \geq \varepsilon\right) \leq 2\exp(-2N_s\varepsilon^2). \tag{19}$$

Since there are $B$ bins, then:

$$
\begin{aligned}
P\left(\max_{1\leq i\leq B}\left|F_{b_i} - P(\hat{S} \in b_i)\right| \geq \varepsilon\right) &= P\left(\bigcup_{i=1}^{B}\left\{\left|F_{b_i} - P(\hat{S} \in b_i)\right|\right\}\right) \\
&\leq \sum_{i=1}^{B} P\left(\left|F_{b_i} - P(\hat{S} \in b_i)\right| \geq \varepsilon\right) \leq 2B\exp(-2N_s\varepsilon^2).
\end{aligned}
\tag{20}
$$

Therefore, $\forall \delta \in (0,1)$, when $N_s \geq \frac{1}{2\varepsilon^2}\ln(\frac{2B}{\delta})$, it holds with probability $1-\delta$:

$$\left|F_{b_i} - P(\hat{S} \in b_i)\right| < \varepsilon, \forall b \in \{b_i\}_{i=1}^{B}. \tag{21}$$

Therefore, if $\sharp D_s^{(k)} \geq \frac{1}{2\varepsilon^2}\ln(\frac{2B}{\delta})$, $\forall b \in \{b_i\}_{i=1}^{B}$, it holds with probability $1-\delta$:

$$
\begin{cases}
|\hat{P}(\hat{S} \in b) - P(\hat{S} \in b)| < \varepsilon, \\
|\hat{P}(\hat{S} \in b|Y=k,\hat{Y}=k) - P(\hat{S} \in b|Y=k,\hat{Y}=k)| < \varepsilon,
\end{cases}
\tag{22}
$$

where $\hat{P}$ represents the estimated values of $P$. Similarly, if $\sharp \hat{D}_t^{(k)} \geq \frac{1}{2\varepsilon^2} \ln(\frac{2B}{\delta})$, $\forall b \in \{b_i\}_{i=1}^B$, it holds with probability $1 - \delta$:

$$\begin{cases} |\hat{Q}(\hat{S} \in b) - Q(\hat{S} \in b)| < \varepsilon, \\ |\hat{Q}(\hat{S} \in b|\hat{Y} = k) - Q(\hat{S} \in b|\hat{Y} = k)| < \varepsilon, \end{cases} \tag{23}$$

where $\hat{Q}$ represents the estimated values of $Q$.

Let $F'_{b_i} = \frac{1}{\sharp b_i} \sum_{j=1}^{\sharp b_i} \mathbf{1}_{\{Y = k \wedge \hat{Y} = k\}}(Y_j, \hat{Y}_j)$, by Hoeffding's inequality, it holds that $\forall \varepsilon > 0$, then:

$$P\left(\left|F'_{b_i} - P(Y = k, \hat{Y} = k|\hat{S} \in b_i)\right| \geq \varepsilon\right) \leq 2\exp(-2\sharp b_i \cdot \varepsilon^2). \tag{24}$$

Since there are $B$ bins, and $\sharp b_i = N_s/B$, and then:

$$\begin{aligned} P\left(\max_{1 \leq i \leq B} \left|F'_{b_i} - P(Y = k, \hat{Y} = k|\hat{S} \in b_i)\right| \geq \varepsilon\right) \\ = P\left(\bigcup_{i=1}^B \left\{\left|F'_{b_i} - P(Y = k, \hat{Y} = k|\hat{S} \in b_i)\right|\right\}\right) \\ \leq \sum_{i=1}^B P\left(\left|F'_{b_i} - P(Y = k, \hat{Y} = k|\hat{S} \in b_i)\right| \geq \varepsilon\right) \leq 2B \exp(-2\frac{N_s}{B}\varepsilon^2). \end{aligned} \tag{25}$$

Therefore, $\forall \delta \in (0, 1)$, when $N_s \geq \frac{B}{2\varepsilon^2} \ln(\frac{2B}{\delta})$, it holds with probability $1 - \delta$:

$$\left|F'_{b_i} - P(Y = k, \hat{Y} = k|\hat{S} \in b_i)\right| < \varepsilon, \forall b \in \{b_i\}_{i=1}^B. \tag{26}$$

Therefore, if $\sharp D_s^{(k)} \geq \frac{B}{2\varepsilon^2} \ln(\frac{2B}{\delta})$, $\forall b \in \{b_i\}_{i=1}^B$, it holds with probability $1 - \delta$:

$$\begin{cases} |\hat{P}(\hat{Y} = k|\hat{S} \in b) - P(\hat{Y} = k|\hat{S} \in b)| < \varepsilon, \\ |\hat{P}(Y = k, \hat{Y} = k|\hat{S} \in b) - P(Y = k, \hat{Y} = k|\hat{S} \in b)| < \varepsilon. \end{cases} \tag{27}$$

Similarly, applying the Hoeffding inequality, we can also obtain: $\forall \delta \in (0, 1)$, when $\sharp D_s^{(k)} \geq \frac{1}{2\varepsilon^2} \ln(\frac{2}{\delta})$, it holds with probability $1 - \delta$:

$$\begin{cases} |\hat{P}(\hat{Y} = k) - P(\hat{Y} = k)| < \varepsilon, \\ |\hat{P}(Y = k, \hat{Y} = k) - P(Y = k, \hat{Y} = k)| < \varepsilon. \end{cases} \tag{28}$$

In summary, if $\min\{\sharp D_s^{(k)}, \sharp \hat{D}_t^{(k)}\} \geq \frac{B}{2\varepsilon^2} \ln(\frac{2B}{\delta})$, $\forall b \in \{b_i\}_{i=1}^B$, it holds with probability $1 - \delta$:

$$\begin{cases} |\hat{P}(\hat{S} \in b) - P(\hat{S} \in b)| < \varepsilon, \\ |\hat{P}(\hat{S} \in b|Y = k, \hat{Y} = k) - P(\hat{S} \in b|Y = k, \hat{Y} = k)| < \varepsilon, \\ |\hat{Q}(\hat{S} \in b) - Q(\hat{S} \in b)| < \varepsilon, \\ |\hat{Q}(\hat{S} \in b|\hat{Y} = k) - Q(\hat{S} \in b|\hat{Y} = k)| < \varepsilon, \\ |\hat{P}(\hat{Y} = k|\hat{S} \in b) - P(\hat{Y} = k|\hat{S} \in b)| < \varepsilon, \\ |\hat{P}(Y = k, \hat{Y} = k|\hat{S} \in b) - P(Y = k, \hat{Y} = k|\hat{S} \in b)| < \varepsilon, \\ |\hat{P}(\hat{Y} = k) - P(\hat{Y} = k)| < \varepsilon, \\ |\hat{P}(Y = k, \hat{Y} = k) - P(Y = k, \hat{Y} = k)| < \varepsilon. \end{cases} \tag{29}$$

Let $V$ represent the vector consisting of all quantities that need to be estimated in Theorem 2's empirical computation, and $\hat{V}$ is the estimated vector of $V$.

**Define the working domain:**

$$D = \left\{\hat{S} \,\Big|\, \hat{Q}(\hat{S} \in b_i) \geq \tau_q \text{ and } \left|\hat{P}(\hat{S} \in b|Y = k, \hat{Y} = k) - \hat{P}(\hat{S} \in b|Y \neq k, \hat{Y} = k)\right| \geq \gamma\right\}, \tag{30}$$

where $\tau_q > 0$ and $\gamma > 0$ are fixed constants. This guarantees all denominators in Eqs. 5–6 are bounded away from 0.

**Lipschitz step on $D$:** On $D$, Eqs. 5 and 6 are Lipschitz continuous with respect to the entries of $V$. Hence there exists $L = L(\tau_q, \gamma) > 0$ such that, if $\|V - \hat{V}\|_\infty \leq \varepsilon/L$, then:

$$\left|\hat{Q}(H = 1|\hat{S} \in b) - Q(H = 1|\hat{S} \in b)\right| \leq \varepsilon, \quad \text{for all } b \in \{b_i\}_{i=1}^B. \tag{31}$$

**Include-pole bins:** If for some $(b, k)$ we have $\left|\hat{P}(\hat{S} \in b|Y = k, \hat{Y} = k) - \hat{P}(\hat{S} \in b|Y \neq k, \hat{Y} = k)\right| < \gamma$ (It is similar for $Q(\hat{S} \in b_i)$), we *interpolate* between the nearest two non-pole bins $b^-, b^+$:

$$\hat{Q}_{\text{int}}\left(H = 1 \mid \hat{S} \in b\right) := (1 - \lambda)\hat{Q}\left(H = 1 \mid \hat{S} \in b^-\right) + \lambda\hat{Q}\left(H = 1 \mid \hat{S} \in b^+\right), \tag{32}$$

where $0 \leq \lambda \leq 1$ weights $b$ between $b^-$ and $b^+$. Let $Q(b)$ represents $Q(H = 1 \mid \hat{S} \in b)$, $\hat{Q}(b^-)$ represents $\hat{Q}\left(H = 1 \mid \hat{S} \in b^-\right)$, and $\hat{Q}(b^+)$ represents $\hat{Q}\left(H = 1 \mid \hat{S} \in b^+\right)$, and then:

$$\begin{aligned}
&\left|\hat{Q}_{\text{int}}\left(H = 1 \mid \hat{S} \in b\right) - Q(H = 1 \mid \hat{S} \in b)\right| \\
&\leq (1 - \lambda)\left|\hat{Q}(b^-) - Q(b^-)\right| + \lambda\left|\hat{Q}(b^+) - Q(b^+)\right| + \left|(1 - \lambda)Q(b^-) + \lambda Q(b^+) - Q(b)\right|.
\end{aligned} \tag{33}$$

The first two terms are $\leq \varepsilon$ as shown above. When $|b^+ - b^-| \leq \varepsilon_b$, due to the continuity of $Q(b)$, it holds that:

$$\left|(1 - \lambda)Q(b^-) + \lambda Q(b^+) - Q(b)\right| \leq \left|Q(b^+) - Q(b^-)\right| \leq L_b \cdot \left|b^+ - b^-\right| \leq L_b \cdot \varepsilon_b, \tag{34}$$

where $L_b$ is a Lipschitz constant. $\qquad\square$

# E   PROOF OF THEOREM 4

$$\begin{aligned}
P(H = 1|\hat{S}, Y = k) &= P(Y = \hat{Y}|\hat{S}, Y = k) \\
&= \frac{P(Y = \hat{Y}, \hat{S}|Y = k)}{P(\hat{S}|Y = k)} = \frac{P(\hat{S}|Y = \hat{Y}, Y = k)P(Y = \hat{Y}|Y = k)}{P(\hat{S}|Y = k)} \\
&= \frac{Q(\hat{S}|\hat{Y} = k, Y = k)Q(\hat{Y} = k|Y = k)}{Q(\hat{S}|Y = k)} = Q(H = 1|\hat{S}, Y = k),
\end{aligned} \tag{35}$$

where the fourth equality is due to Eq. 14 and Definition 3.

# F   PROOF OF THEOREM 5

$$\begin{aligned}
P(H = 1|\hat{S}) &= P(H = 1, Y = 0|\hat{S}) + P(H = 1, Y = 1|\hat{S}) \\
&= P(H = 1|\hat{S}, Y = 0)P(Y = 0|\hat{S}) + P(H = 1|\hat{S}, Y = 1)P(Y = 1|\hat{S}) \\
&= P(H = 1|\hat{S}, Y = 0)\frac{P(\hat{S}|Y = 0)P(Y = 0)}{P(\hat{S})} + P(H = 1|\hat{S}, Y = 1)\frac{P(\hat{S}|Y = 1)P(Y = 1)}{P(\hat{S})} \\
&= P(H = 1|\hat{S}, Y = 0)\frac{P(\hat{S}|Y = 0)P(Y = 0)}{P(\hat{S}|Y = 0)P(Y = 0) + P(\hat{S}|Y = 1)P(Y = 1)} \\
&+ P(H = 1|\hat{S}, Y = 1)\frac{P(\hat{S}|Y = 1)P(Y = 1)}{P(\hat{S}|Y = 0)P(Y = 0) + P(\hat{S}|Y = 1)P(Y = 1)}.
\end{aligned} \tag{36}$$

Except for the label distribution $P(Y)$, the right side of Eq. 36 only contains preset unchanged quantities between source domain and target domain, so the preset true calibration curve on the target domain can be derived as follows:

$$\begin{aligned}
Q(H = 1|\hat{S}) &= P(H = 1|\hat{S}, Y = 0)\frac{P(\hat{S}|Y = 0)Q(Y = 0)}{P(\hat{S}|Y = 0)Q(Y = 0) + P(\hat{S}|Y = 1)Q(Y = 1)} \\
&+ P(H = 1|\hat{S}, Y = 1)\frac{P(\hat{S}|Y = 1)Q(Y = 1)}{P(\hat{S}|Y = 0)Q(Y = 0) + P(\hat{S}|Y = 1)Q(Y = 1)}.
\end{aligned} \tag{37}$$

---

**Algorithm 1** Target Label-Free Confidence Calibration.

---

1: **Initialize:**
2:    $D_s = \{(\hat{s}_i, y_i, \hat{y}_i)\}_{1 \le i \le N_s}^{N_s}$,
3:    $D_t = \{(\hat{s}_i, \hat{y}_i)\}_{1 \le i \le N_t}^{N_t}$,
4:    **for** $1 \le k \le K$:
5:      $D_s^{(k)} = \{(\hat{s}, y, \hat{y}) | y = k, \hat{y} = k, (\hat{s}, y, \hat{y}) \in D_s\}$,
6:      $\hat{D}_t^{(k)} = \{(\hat{s}, \hat{y}) | \hat{y} = k, (\hat{s}, \hat{y}) \in D_t\}$,
7:    $\{b_i\}_{i=1}^B$, $K$.
8: **Estimating**:
9:    **for** $i \le B$:
10:      Estimate $Q(\hat{S} \in b_i)$ via $D_t$.
11:      Estimate $P(\hat{S} \in b_i)$ via $D_s$.
12:      **for** $1 \le k \le K$:
13:        Estimate $P(\hat{S} \in b_i | Y = k, \hat{Y} = k)$ via $D_s^{(k)}$,
14:        Estimate $Q(\hat{S} \in b_i | \hat{Y} = k)$ via $\hat{D}_t^{(k)}$,
15:        Estimate $P(\hat{Y} = k | \hat{S} \in b)$ via $D_s$,
16:        Estimate $P(Y = k, \hat{Y} = k | \hat{S} \in b_i)$ via $D_s$,
17:    **for** $1 \le k \le K$:
18:      $Q(\hat{Y} = k) = \sharp \hat{D}_t^{(k)} / \sharp D_t$.
19:      Estimate $P(\hat{Y} = k)$ via $D_s$.
20:      Estimate $P(\hat{Y} = k, Y = k)$ via $D_s$.
21:    Compute $P(\hat{S} \in b | Y \ne k, \hat{Y} = k)$ via Eq. 7.
22:    Compute $Q(H = 1 | \hat{Y} = k)$ via Eq. 6.
23: **Calibrating**:
24:    $D_{cali} = \{\}$.
25:    **for** $i \le B$:
26:      Compute $Q(H = 1 | \hat{S} \in b_i)$ using Eq. 5.
27:      Add (mean($b_i$), $Q(H = 1 | \hat{S} \in b_i)$) into $D_{cali}$.
28:    Fit $D_{cali}$ to get $Q(H = 1 | \hat{S})$.
29: **Return** $Q(H = 1 | \hat{S})$.

---

## G  PSEUDO-CODE

**Target Label-Free Confidence Calibration:** Algorithm 1 shows Theorem 2's empirical computation process. Since the target domain dataset $D_t$ does not contain the true labels, the proposed method is an unsupervised domain adaptation method. The estimating step (Line 8 in Algorithm 1) estimates the required probabilities in an unbiased manner through approximating probabilities by frequency. Calibrating step (Line 23 in Algorithm 1) uses the estimated probabilities to obtain calibration results. In practical, to improve the calibration robustness, calibration results under multiple binning strategies can be collected and then fit all the calibration results. In all experiments in this paper, the fitting operation (i.e., line 28 of Algorithm 1) uses generalized linear models (GLM) and selects the best model by using the Akaike Information Criteria (AIC).

**Simulation Data Generation Method:** Algorithm 2 shows the method of generating realistic source domain and target domain simulation data, where the sampling method of $\hat{S}$ (i.e., line 11 and line 18) adopts beta distribution sampling and the sampling method $\text{BI}(\cdot)$ of $H$(i.e., line 12 and line 19) adopts the binomial distribution sampling proposed by Dong et al. (2025b). The settings of $g_1(\hat{S})$ and $g_2(\hat{S})$ can adopt the functions in Table 4. The generated simulation data can be used as a benchmark to evaluate the effectiveness of the calibration curve estimation method, as shown in Section 5.1.

---

**Algorithm 2** Simulation Data Generation Method.

---

1: **Initialize:**
2:     $N_s$, $N_t$, $P(Y = 0)$, $Q(Y = 0)$,
3:     $\hat{S}|Y = 0 \sim \text{Be}(\alpha_1, \beta_1)$, $\hat{S}|Y = 1 \sim \text{Be}(\alpha_2, \beta_2)$,
4:     $P(H = 1|\hat{S}, Y = 0) = g_1(\hat{S})$,
5:     $P(H = 1|\hat{S}, Y = 1) = g_2(\hat{S})$.
6: **Sampling:**
7:     **for** $(N,p)$ in $\{(N_s, P(Y = 0)), (N_t, Q(Y = 0))\}$:
8:         $i = 1$; $D = \{\}$.
9:         **while** $i \leq N$:
10:             **if** random() $\leq p$:
11:                 $\hat{s}$ = Sampling from $\text{Be}(\alpha_1, \beta_1)$,
12:                 $H$ = Sampling from $\text{BI}(1, g_1(\hat{S}))$,
13:                 **if** $H == 1$:
14:                     $y = 0$; $\hat{y} = 0$.
15:                 **else**:
16:                     $y = 0$; $\hat{y} = 1$.
17:             **else**:
18:                 $\hat{s}$ = Sampling from $\text{Be}(\alpha_2, \beta_2)$,
19:                 $H$ = Sampling from $\text{BI}(1, g_2(\hat{S}))$,
20:                 **if** $H == 1$:
21:                     $y = 1$; $\hat{y} = 1$.
22:                 **else**:
23:                     $y = 1$; $\hat{y} = 0$.
24:             Add $(\hat{s}, y, \hat{y})$ into $D$.
25:         **if** $p == P(Y = 0)$:
26:             $D_s = D$.
27:         **elif** $p == Q(Y = 0)$:
28:             $D_t = D$.
29: **Return** $D_s$, $D_t$.

---

## H RESULTS

### H.1 OTHER EXPERIMENTAL SETUP

#### H.1.1 SOFTWARE AND HARDWARE ENVIRONMENT AND HYPERPARAMETERS

All experiment was conducted on Intel® Core$^{TM}$ I7-10700 CPU with 3.70GHz and 125.5GB memory, NVIDIA Quadro RTX 5000 graphics card with 16GB of video memory, Ubuntu 20.04.3 LTS, Python 3.8.12, and Torch 2.3.1+cu118. We use the SGD optimizer to train classifier for 150 epochs, with an initial learning rate of 0.01 and a learning rate of 0.001 from epoch 75 to 150. The batch size for all training is 128. The number of bins for calibration metrics that require binning is set to the popular 15 (Guo et al., 2017; Dong et al., 2025b; Roelofs et al., 2022).

#### H.1.2 TRUE DISTRIBUTION'S PRESET

When simulating the label shift dataset, the true class-condition confidence distribution $\hat{S}|Y = k$ and the true class-condition calibration curve $P(H = 1|\hat{S}, Y = k)$ need to be preset, as shown in Section 4. Referring to the fitting results of Roelofs et al. (2022) and the preset schemes of Dong et al. (2025b), we select six preset schemes, as shown in Table 4. Fig. 4 shows a visualization of true class-condition calibration curves of three schemes, including various degrees of curve differences, e.g., the curve difference in D2 is small, and the curve difference in D6 is large.

#### H.1.3 PROCESSING OF REAL-WORLD DATASETS

German Credit dataset and Dry Bean dataset both are class imbalanced, and the following processing is performed to achieve label shift: the test set is sampled to balance the data, and the remaining

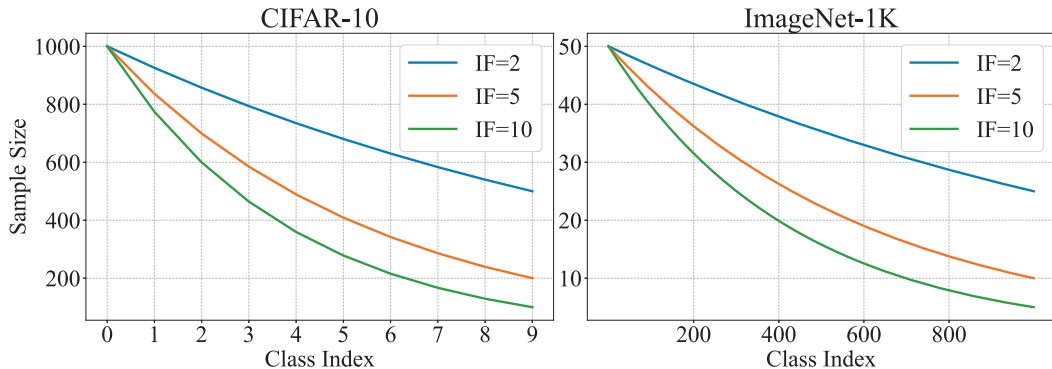

Figure 3: Sample size per class in the target domain in simulated long-tail SVHN/CIFAR-10/ImageNet-1K with different imbalance factors (IF).

imbalanced data is the training set. MHIST, SVHN, CIFAR-10, CIFAR-100, and ImageNet-1K are resampled into label shift datasets, and the resampled datasets are named MHIST-LS, SVHN-LS, CIFAR-10-LS, and ImageNet-LS. In the MHIST-LS dataset, label shift is achieved by controlling the sample ratio of $Y = 0$ in the source and target domains. In SVHN-LS, CIFAR-10-LS, CIFAR-100-LS, and ImageNet-LS, the source domains are uniformly distributed, and the target domains are resampled into long-tailed distributions through a imbalance factor (IF) (Popordanoska et al., 2024). Specifically, IF controls the ratio between the sample size in the most frequent and the least frequent class. For example, an imbalance factor of 10 indicates that the least frequent class appears 10 times less than the most frequent one. Fig. 3 shows sample size per class in the target domain in simulated long-tail CIFAR-10/ImageNet-1K with different imbalance factors.

### H.1.4 SELECTION OF NEURAL NETWORK CLASSIFIERS

The commonly used networks on these datasets are used in the experiments, i.e., LeNet-1D (Lecun et al., 1998), MLP (Bishop, 1995), and TabNet (Arik & Pfister, 2021) for German Credit and Dry Bean data , ResNet (He et al., 2016) for MHIST-LS, SVHN-LS, CIFAR-10-LS, and CIFAR-100-LS, and Wide-ResNet (Zagoruyko & Komodakis, 2016), DenseNet-162 (Huang et al., 2017), and ViT (Dosovitskiy et al., 2021) for ImageNet-LS.

Table 4: Selection of preset schemes, where logflip $= \log(1 - x)$.

| Name | Class | $\mathbf{P(H = 1|\hat{S}, Y = k)}$ | $\hat{\mathbf{S}}|\mathbf{Y = k}$ |
|------|-------|-----------------------------------|-----------------------------------|
| D1 | Y=0 | $\text{logit}^{-1}(-0.88 + 0.49 \cdot \text{logit}(\hat{S}))$ | Be(1.12, 0.11) |
|    | Y=1 | $\text{logflip}^{-1}(-0.12 + 0.58 \cdot \text{logflip}(\hat{S}))$ | Be(2.17, 0.03) |
| D2 | Y=0 | $\log^{-1}(-0.03 + 1.27 \cdot \log(\hat{S}))$ | Be(1.13, 0.20) |
|    | Y=1 | $\text{logit}^{-1}(-0.77 - 0.80 \cdot \text{logflip}(\hat{S}))$ | Be(1.19, 0.22) |
| D3 | Y=0 | $\log^{-1}(-0.03 + 1.27 \cdot \log(\hat{S}))$ | Be(1.17, 0.15) |
|    | Y=1 | $\text{logit}^{-1}(-0.97 + 0.34 \cdot \text{logit}(\hat{S}))$ | Be(2.19, 0.35) |
| D4 | Y=0 | $\log^{-1}(-0.05 + 2.52 \cdot \log(\hat{S}))$ | Be(1.92, 0.13) |
|    | Y=1 | $\text{logflip}^{-1}(-0.20 + 0.70 \cdot \text{logflip}(\hat{S}))$ | Be(1.19, 0.14) |
| D5 | Y=0 | $\log^{-1}(-0.02 + 2.12 \cdot \log(\hat{S}))$ | Be(1.83, 0.10) |
|    | Y=1 | $\text{logflip}^{-1}(-0.20 + 0.75 \cdot \text{logflip}(\hat{S}))$ | Be(2.05, 0.40) |
| D6 | Y=0 | $\text{logit}^{-1}(-0.90 + 0.56 \cdot \text{logit}(\hat{S}))$ | Be(1.53, 0.10) |
|    | Y=1 | $\text{logit}^{-1}(-0.55 - 0.90 \cdot \text{logflip}(\hat{S}))$ | Be(1.35, 0.20) |

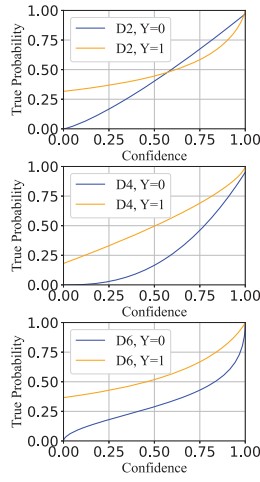

Figure 4: Show class-condition curves.

## H.2 OTHER RESULTS

Table 5: Compare calibration errors on real-world data. "Res" is ResNet (He et al., 2016), "W-Res" is Wide-ResNet (Zagoruyko & Komodakis, 2016), "Dense" is DenseNet (Huang et al., 2017), and "ViT-L" is ViT-Large (Dosovitskiy et al., 2021). "0.8→0.4" indicates $P(Y = 0) = 0.8$ and $Q(Y = 0) = 0.4$. The reported results are mean ± std over ten runs.

| Dataset | $\mathbf{ECE_{debiased}} \downarrow$ | | | | | | |
| --- | --- | --- | --- | --- | --- | --- | --- |
| | Uncal | TempScal | PCS | LADE | MRR | LaSCal | TLFCC |
| ***German Credit*** | | | | | | | |
| LeNet-1D | $37.03_{\pm 1.82}$ | $12.11_{\pm 0.33}$ | $6.481_{\pm 0.28}$ | $7.742_{\pm 0.28}$ | $8.331_{\pm 0.41}$ | $5.834_{\pm 0.26}$ | $5.195_{\pm 0.21}$ |
| MLP | $27.99_{\pm 1.35}$ | $14.03_{\pm 0.55}$ | $9.182_{\pm 0.27}$ | $13.84_{\pm 0.41}$ | $11.69_{\pm 0.40}$ | $7.239_{\pm 0.35}$ | $5.121_{\pm 0.15}$ |
| TabNet | $37.22_{\pm 1.72}$ | $12.01_{\pm 0.38}$ | $10.50_{\pm 0.40}$ | $5.957_{\pm 0.28}$ | $8.348_{\pm 0.31}$ | $5.768_{\pm 0.18}$ | $5.187_{\pm 0.20}$ |
| ***Dry Bean*** | | | | | | | |
| LeNet-1D | $64.36_{\pm 2.50}$ | $42.92_{\pm 1.11}$ | $38.86_{\pm 1.61}$ | $7.201_{\pm 0.21}$ | $7.596_{\pm 0.28}$ | $0.559_{\pm 0.02}$ | $0.258_{\pm 0.01}$ |
| MLP | $63.82_{\pm 2.46}$ | $41.87_{\pm 1.89}$ | $18.02_{\pm 0.70}$ | $8.461_{\pm 0.29}$ | $8.759_{\pm 0.41}$ | $0.587_{\pm 0.01}$ | $0.505_{\pm 0.02}$ |
| TabNet | $64.89_{\pm 1.99}$ | $50.46_{\pm 1.96}$ | $45.47_{\pm 1.93}$ | $19.56_{\pm 0.61}$ | $6.935_{\pm 0.32}$ | $1.111_{\pm 0.04}$ | $0.566_{\pm 0.02}$ |
| ***MHIST-LS*** | | | | | | | |
| Res18 (0.8→0.4) | $22.76_{\pm 0.86}$ | $11.17_{\pm 0.37}$ | $5.121_{\pm 0.18}$ | $7.117_{\pm 0.24}$ | $6.515_{\pm 0.21}$ | $4.821_{\pm 0.23}$ | $4.315_{\pm 0.20}$ |
| Res50 (0.7→0.4) | $24.35_{\pm 1.09}$ | $8.203_{\pm 0.35}$ | $6.363_{\pm 0.18}$ | $4.066_{\pm 0.12}$ | $2.938_{\pm 0.10}$ | $2.170_{\pm 0.08}$ | $2.077_{\pm 0.10}$ |
| Res101 (0.9→0.3) | $26.10_{\pm 0.88}$ | $3.210_{\pm 0.13}$ | $2.469_{\pm 0.07}$ | $2.558_{\pm 0.07}$ | $1.094_{\pm 0.05}$ | $1.021_{\pm 0.03}$ | $0.242_{\pm 0.01}$ |
| ***SVHN-LS*** | | | | | | | |
| Res20 (IF = 2) | $34.50_{\pm 1.72}$ | $16.42_{\pm 0.69}$ | $10.30_{\pm 0.42}$ | $14.95_{\pm 0.48}$ | $6.210_{\pm 0.26}$ | $1.640_{\pm 0.05}$ | $0.910_{\pm 0.03}$ |
| Res56 (IF = 5) | $31.20_{\pm 1.60}$ | $14.90_{\pm 0.65}$ | $9.211_{\pm 0.39}$ | $13.02_{\pm 0.45}$ | $5.740_{\pm 0.24}$ | $1.520_{\pm 0.05}$ | $0.840_{\pm 0.03}$ |
| Res110 (IF = 10) | $29.40_{\pm 1.55}$ | $13.88_{\pm 0.63}$ | $8.760_{\pm 0.37}$ | $12.11_{\pm 0.43}$ | $5.320_{\pm 0.23}$ | $1.460_{\pm 0.04}$ | $0.780_{\pm 0.02}$ |
| ***CIFAR-10-LS*** | | | | | | | |
| Res20 (IF = 2) | $72.59_{\pm 2.79}$ | $36.50_{\pm 1.69}$ | $4.613_{\pm 0.21}$ | $6.530_{\pm 0.19}$ | $6.809_{\pm 0.17}$ | $0.882_{\pm 0.02}$ | $0.456_{\pm 0.01}$ |
| Res56 (IF = 5) | $65.61_{\pm 2.07}$ | $43.62_{\pm 1.62}$ | $5.887_{\pm 0.23}$ | $10.12_{\pm 0.31}$ | $10.93_{\pm 0.28}$ | $3.538_{\pm 0.11}$ | $0.570_{\pm 0.02}$ |
| Res110 (IF = 10) | $71.53_{\pm 2.47}$ | $27.82_{\pm 1.13}$ | $4.965_{\pm 0.14}$ | $22.34_{\pm 0.57}$ | $9.160_{\pm 0.29}$ | $2.462_{\pm 0.06}$ | $0.974_{\pm 0.04}$ |
| ***CIFAR-100-LS*** | | | | | | | |
| Res20 (IF = 2) | $73.80_{\pm 2.90}$ | $44.21_{\pm 1.96}$ | $26.55_{\pm 1.07}$ | $45.63_{\pm 1.82}$ | $22.30_{\pm 1.02}$ | $8.140_{\pm 0.29}$ | $6.910_{\pm 0.23}$ |
| Res56 (IF = 5) | $69.50_{\pm 2.74}$ | $42.08_{\pm 1.88}$ | $24.33_{\pm 1.01}$ | $42.74_{\pm 1.72}$ | $20.85_{\pm 0.98}$ | $7.760_{\pm 0.27}$ | $6.580_{\pm 0.22}$ |
| Res110 (IF = 10) | $71.20_{\pm 2.78}$ | $42.95_{\pm 1.91}$ | $25.10_{\pm 1.03}$ | $43.80_{\pm 1.74}$ | $21.40_{\pm 1.00}$ | $7.890_{\pm 0.28}$ | $6.660_{\pm 0.22}$ |
| ***ImageNet-LS*** | | | | | | | |
| W-Res50 (IF=2) | $59.38_{\pm 1.49}$ | $39.33_{\pm 1.17}$ | $27.74_{\pm 1.01}$ | $33.67_{\pm 1.48}$ | $39.20_{\pm 1.27}$ | $23.31_{\pm 0.67}$ | $7.898_{\pm 0.25}$ |
| Dense162 (IF=5) | $82.68_{\pm 3.17}$ | $55.53_{\pm 1.74}$ | $29.16_{\pm 0.94}$ | $47.60_{\pm 2.10}$ | $23.07_{\pm 1.11}$ | $7.929_{\pm 0.37}$ | $7.659_{\pm 0.29}$ |
| ViT-L (IF=10) | $78.40_{\pm 3.86}$ | $63.25_{\pm 1.96}$ | $22.00_{\pm 0.84}$ | $54.13_{\pm 2.35}$ | $28.95_{\pm 1.19}$ | $10.71_{\pm 0.51}$ | $7.095_{\pm 0.32}$ |

**Additional Results on $ECE_{debiased}$ and KS-error:** Table 5 and Table 6 further confirm the effectiveness of TLFCC across alternative calibration metrics. For $ECE_{debiased}$, TLFCC consistently achieves the lowest error on all datasets, reducing calibration error by large margins compared to LaSCal (e.g., from 1.021% to 0.242% on MHIST-LS Res101 and from 10.71% to 7.095% on ImageNet-LS ViT-L). Similar trends are observed for KS-error, where TLFCC maintains clear superiority under severe label shift, such as CIFAR-10-LS with an imbalance factor of 10, achieving 0.610% versus LaSCal's 2.299%. These results demonstrate that the proposed method not only excels under the standard ECE metric but also generalizes well to stricter statistical measures, reinforcing its effectiveness and stability in real-world label-shift scenarios.

Fig. 5 illustrates the comparative reliability diagrams for three methods— Uncal, LaSCal, and TLFCC—across MHIST-LT, CIFAR-100-LT, and ImageNet-LT. The diagrams clearly show that TLFCC achieves the smallest gap between predicted confidence and empirical accuracy, resulting in the lowest ECE values among all methods. While LaSCal substantially improves calibration over the uncalibrated baseline, its residual gap remains noticeable, especially under severe label shift as seen in CIFAR-100-LT and ImageNet-LT. In contrast, TLFCC consistently produces curves that closely follow the diagonal, indicating near-perfect calibration. These results confirm that TLFCC

Table 6: Compare calibration errors on real-world data. "Res" is ResNet (He et al., 2016), "W-Res" is Wide-ResNet (Zagoruyko & Komodakis, 2016), "Dense" is DenseNet (Huang et al., 2017), and "ViT-L" is ViT-Large (Dosovitskiy et al., 2021). "0.8→0.4" indicates $P(Y = 0) = 0.8$ and $Q(Y = 0) = 0.4$. The reported results are mean ± std over ten runs.

| Dataset | KS-error ↓ | | | | | | |
|---|---|---|---|---|---|---|---|
| | Uncal | TempScal | PCS | LADE | MRR | LaSCal | TLFCC |
| **_German Credit_** | | | | | | | |
| LeNet-1D | $23.64_{\pm0.68}$ | $11.21_{\pm0.48}$ | $10.53_{\pm0.48}$ | $4.559_{\pm0.13}$ | $5.436_{\pm0.22}$ | $4.474_{\pm0.19}$ | $3.771_{\pm0.11}$ |
| MLP | $28.47_{\pm0.79}$ | $13.31_{\pm0.49}$ | $8.177_{\pm0.33}$ | $11.85_{\pm0.30}$ | $12.39_{\pm0.50}$ | $7.468_{\pm0.19}$ | $5.161_{\pm0.21}$ |
| TabNet | $23.64_{\pm0.68}$ | $11.21_{\pm0.30}$ | $10.97_{\pm0.34}$ | $8.096_{\pm0.37}$ | $5.436_{\pm0.25}$ | $4.474_{\pm0.17}$ | $3.771_{\pm0.18}$ |
| **_Dry Bean_** | | | | | | | |
| LeNet-1D | $64.33_{\pm1.77}$ | $42.93_{\pm1.62}$ | $31.94_{\pm1.44}$ | $5.770_{\pm0.27}$ | $3.540_{\pm0.09}$ | $0.761_{\pm0.03}$ | $0.348_{\pm0.01}$ |
| MLP | $63.82_{\pm2.35}$ | $41.88_{\pm1.14}$ | $34.61_{\pm1.63}$ | $2.397_{\pm0.06}$ | $4.544_{\pm0.22}$ | $0.637_{\pm0.02}$ | $0.295_{\pm0.01}$ |
| TabNet | $64.88_{\pm2.42}$ | $50.45_{\pm1.29}$ | $50.16_{\pm2.06}$ | $9.603_{\pm0.47}$ | $3.051_{\pm0.11}$ | $1.344_{\pm0.03}$ | $0.944_{\pm0.04}$ |
| **_MHIST-LS_** | | | | | | | |
| Res18 (0.8→0.4) | $23.07_{\pm0.77}$ | $6.546_{\pm0.32}$ | $5.725_{\pm0.16}$ | $5.070_{\pm0.14}$ | $5.290_{\pm0.18}$ | $4.947_{\pm0.20}$ | $4.435_{\pm0.14}$ |
| Res50 (0.7→0.4) | $24.40_{\pm0.74}$ | $3.209_{\pm0.11}$ | $1.211_{\pm0.05}$ | $2.891_{\pm0.13}$ | $3.109_{\pm0.10}$ | $1.190_{\pm0.03}$ | $0.818_{\pm0.03}$ |
| Res101 (0.9→0.3) | $26.10_{\pm1.10}$ | $1.590_{\pm0.04}$ | $1.092_{\pm0.05}$ | $1.079_{\pm0.03}$ | $1.126_{\pm0.05}$ | $1.047_{\pm0.03}$ | $0.349_{\pm0.01}$ |
| **_SVHN-LS_** | | | | | | | |
| Res20 (IF = 2) | $22.80_{\pm1.30}$ | $11.35_{\pm0.52}$ | $7.420_{\pm0.33}$ | $9.860_{\pm0.38}$ | $4.310_{\pm0.19}$ | $1.180_{\pm0.04}$ | $0.590_{\pm0.02}$ |
| Res56 (IF = 5) | $21.10_{\pm1.25}$ | $10.72_{\pm0.50}$ | $6.880_{\pm0.31}$ | $9.140_{\pm0.36}$ | $4.080_{\pm0.18}$ | $1.120_{\pm0.04}$ | $0.550_{\pm0.02}$ |
| Res110 (IF = 10) | $19.90_{\pm1.22}$ | $10.11_{\pm0.49}$ | $6.540_{\pm0.30}$ | $8.710_{\pm0.35}$ | $3.960_{\pm0.17}$ | $1.080_{\pm0.03}$ | $0.520_{\pm0.02}$ |
| **_CIFAR-10-LS_** | | | | | | | |
| Res20 (IF = 2) | $72.59_{\pm3.53}$ | $36.50_{\pm1.69}$ | $10.53_{\pm0.38}$ | $11.13_{\pm0.53}$ | $3.160_{\pm0.08}$ | $0.616_{\pm0.01}$ | $0.538_{\pm0.01}$ |
| Res56 (IF = 5) | $65.60_{\pm1.84}$ | $43.62_{\pm1.32}$ | $30.19_{\pm0.97}$ | $5.204_{\pm0.17}$ | $4.128_{\pm0.17}$ | $3.542_{\pm0.12}$ | $0.565_{\pm0.01}$ |
| Res110 (IF = 10) | $71.54_{\pm2.52}$ | $27.81_{\pm1.10}$ | $4.232_{\pm0.16}$ | $26.30_{\pm0.92}$ | $3.982_{\pm0.10}$ | $2.299_{\pm0.08}$ | $0.610_{\pm0.01}$ |
| **_CIFAR-100-LS_** | | | | | | | |
| Res20 (IF = 2) | $58.30_{\pm2.25}$ | $33.42_{\pm1.52}$ | $19.80_{\pm0.91}$ | $31.25_{\pm1.28}$ | $15.90_{\pm0.73}$ | $6.210_{\pm0.22}$ | $5.120_{\pm0.17}$ |
| Res56 (IF = 5) | $55.10_{\pm2.18}$ | $31.95_{\pm1.47}$ | $18.44_{\pm0.88}$ | $29.62_{\pm1.22}$ | $15.10_{\pm0.69}$ | $5.980_{\pm0.21}$ | $4.960_{\pm0.16}$ |
| Res110 (IF = 10) | $56.40_{\pm2.20}$ | $32.40_{\pm1.49}$ | $18.95_{\pm0.89}$ | $30.10_{\pm1.24}$ | $15.40_{\pm0.70}$ | $6.040_{\pm0.21}$ | $5.000_{\pm0.16}$ |
| **_ImageNet-LS_** | | | | | | | |
| W-Res50 (IF=2) | $59.38_{\pm2.73}$ | $39.32_{\pm1.61}$ | $23.33_{\pm1.01}$ | $36.92_{\pm1.73}$ | $23.31_{\pm0.81}$ | $18.65_{\pm0.84}$ | $3.747_{\pm0.15}$ |
| Dense162 (IF=5) | $82.68_{\pm3.52}$ | $55.52_{\pm2.69}$ | $21.18_{\pm0.77}$ | $11.20_{\pm0.36}$ | $12.31_{\pm0.60}$ | $7.941_{\pm0.39}$ | $7.669_{\pm0.30}$ |
| ViT-L (IF=10) | $78.40_{\pm3.50}$ | $63.25_{\pm2.49}$ | $44.25_{\pm1.50}$ | $13.45_{\pm0.64}$ | $14.68_{\pm0.38}$ | $10.72_{\pm0.42}$ | $5.056_{\pm0.15}$ |

not only reduces overall calibration error but also maintains robustness across diverse datasets and network architectures, validating its effectiveness as a label-free solution under label shift.

### H.3 OTHER ABLATION EXPERIMENT RESULTS

#### H.3.1 IMPACT OF ESTIMATION METHODS

Table 7 and Table 8 show that different estimation strategies for confidence distribution (Beta vs. histogram binning) and calibration curve fitting (cubic smoothing spline vs. generalized linear models) have only a marginal effect on TLFCC's performance. Across all tested combinations—such as Beta distribution versus histogram binning for density estimation and cubic smoothing spline versus generalized linear models for curve fitting—the variation in ECE remains minimal (within 2%). This consistency demonstrates that TLFCC is highly robust to the choice of estimation method, ensuring stable calibration performance.

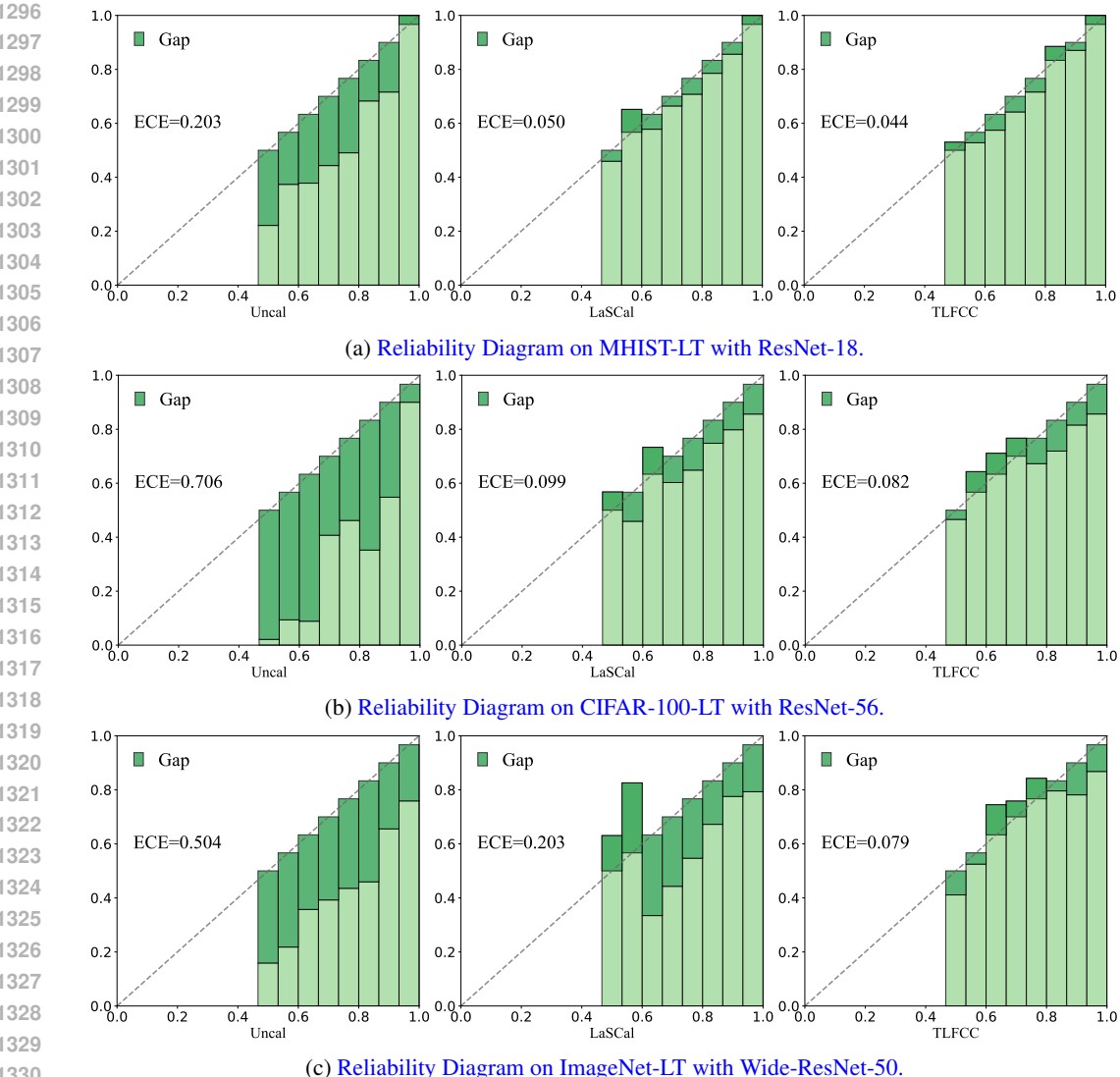

(a) Reliability Diagram on MHIST-LT with ResNet-18.

(b) Reliability Diagram on CIFAR-100-LT with ResNet-56.

(c) Reliability Diagram on ImageNet-LT with Wide-ResNet-50.

Figure 5: Reliability diagrams on real-world data.

Table 7: Impact of Estimation Methods. The dataset is CIFAR-10-LS (IF=10) and the classifier is ResNet56.

| BETA | HB | CSS | GLM | $ECE_{bin}$ (%) $\downarrow$ |
|------|-----|-----|-----|------|
| ✓ | | ✓ | | $0.942_{0.06}$ |
| ✓ | | | ✓ | $0.934_{0.05}$ |
| | ✓ | ✓ | | $0.933_{0.06}$ |
| | ✓ | | ✓ | $0.921_{0.04}$ |

Table 8: Impact of Estimation Methods. The dataset is ImageNet-LS (IF=10) and the classifier is ViT-L.

| BETA | HB | CSS | GLM | $ECE_{bin}$ (%) $\downarrow$ |
|------|-----|-----|-----|------|
| ✓ | | ✓ | | $7.213_{0.39}$ |
| ✓ | | | ✓ | $7.107_{0.38}$ |
| | ✓ | ✓ | | $7.103_{0.30}$ |
| | ✓ | | ✓ | $7.090_{0.28}$ |

### H.3.2 IMPACT OF SHIFT MAGNITUDE

Table 9 and Table 10 show that calibration error consistently increases as the label shift becomes more severe, highlighting the challenge of maintaining reliable confidence estimates under extreme imbalance. All methods degrade with larger imbalance factors. In contrast, TLFCC demonstrates the smallest increase across all scenarios, maintaining low ECE even when the imbalance factor reaches 100. These results confirm that TLFCC offers superior robustness and scalability under varying

shift magnitudes, making it particularly suitable for real-world applications with unpredictable label distribution shifts.

Table 9: Impact of Shift Magnitude. IF refers to the imbalance factor. The dataset is CIFAR-10-LS and the classifier is ResNet56.

Table 10: Impact of Shift Magnitude. IF refers to the imbalance factor. The dataset is ImageNet-LS and the classifier is ViT-L.

| Magnitude | $ECE_{bin}$ (%) $\downarrow$ | | | |
| --- | --- | --- | --- | --- |
| | LADE | MRR | LaSCal | TLFCC |
| IF=2 | $35.0_{1.22}$ | $14.1_{0.44}$ | $3.49_{0.09}$ | $0.77_{0.02}$ |
| IF=5 | $35.1_{1.20}$ | $14.4_{0.44}$ | $3.54_{0.09}$ | $0.78_{0.02}$ |
| IF=10 | $35.9_{1.21}$ | $15.1_{0.46}$ | $3.80_{0.10}$ | $0.92_{0.04}$ |
| IF=25 | $36.2_{1.25}$ | $15.6_{0.45}$ | $4.50_{0.12}$ | $1.05_{0.05}$ |
| IF=50 | $38.7_{1.30}$ | $17.2_{0.53}$ | $6.04_{0.12}$ | $2.11_{0.08}$ |
| IF=100 | $43.6_{1.41}$ | $19.8_{0.61}$ | $8.32_{0.15}$ | $3.04_{0.09}$ |

| Magnitude | $ECE_{bin}$ (%) $\downarrow$ | | | |
| --- | --- | --- | --- | --- |
| | LADE | MRR | LaSCal | TLFCC |
| IF=2 | $22.4_{0.71}$ | $22.0_{0.75}$ | $10.4_{0.32}$ | $7.10_{0.27}$ |
| IF=5 | $22.4_{0.69}$ | $22.2_{0.70}$ | $10.5_{0.33}$ | $7.10_{0.28}$ |
| IF=10 | $22.8_{0.60}$ | $22.2_{0.98}$ | $10.7_{0.33}$ | $7.09_{0.28}$ |
| IF=25 | $24.5_{0.85}$ | $23.1_{0.87}$ | $12.2_{0.35}$ | $7.54_{0.31}$ |
| IF = 50 | $26.2_{0.82}$ | $24.3_{0.80}$ | $13.4_{0.40}$ | $8.31_{0.35}$ |
| IF = 100 | $29.5_{0.91}$ | $27.4_{0.95}$ | $15.3_{0.41}$ | $10.1_{0.37}$ |

### H.3.3 IMPACT OF SAMPLE SIZE

**Impact of Target Domain Sample Size:** As shown in Fig. 6, increasing the number of unlabeled target samples significantly improves calibration stability. With only a few hundred samples, TLFCC already produces a curve close to the true target-domain calibration curve, and further enlarging $N_t$ reduces variance and oscillations. This trend aligns with Theorem 3, confirming that more target data steadily enhances calibration accuracy without requiring target domain labels.

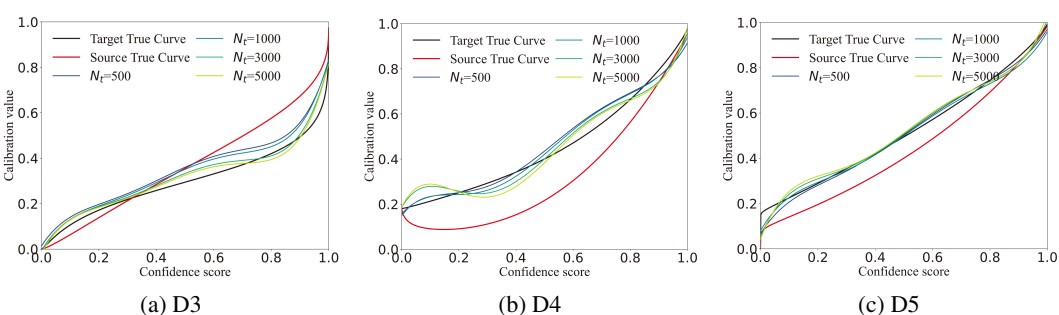

(a) D3  (b) D4  (c) D5

Figure 6: Impact of Target Domain Sample Size.

**Impact of Source Domain Sample Size:** Similar to the effect of increasing target-domain samples, having more labeled data in the source domain significantly improves the stability and accuracy of TLFCC, as shown in Fig. 7. When the source dataset is small, the estimates of key probabilities are noisy, and this noise can lead to fluctuations in the calibration curve. As the source sample size grows, these estimates become more reliable, resulting in smoother calibration curves and fewer extreme deviations. Larger source datasets also reduce the risk of instability in the calculation steps, which can otherwise occur when the difference between correct and incorrect prediction distributions is very small. Therefore, in practice, adding more source samples is particularly helpful under severe label shift.

### H.3.4 POLE ANALYSIS

Fig. 8 illustrates the pole behavior of Theorem 2, which arises when the denominator in Eq. 6 approaches zero, i.e., when $Q(\hat{S} \mid Y = k, \hat{Y} = k) = Q(\hat{S} \mid Y \neq k, \hat{Y} = k)$. In such cases, the estimation of $Q(H = 1 \mid \hat{Y} = k)$ becomes unstable, leading to large fluctuations in the calibration curve. However, from the estimated curves in Fig. 8, there is usually only one such pole, as shown by the intersection point in Fig. 8. Therefore, we can usually safely perform the calibration of Theorem 2 on non-pole regions and interpolate at the poles.

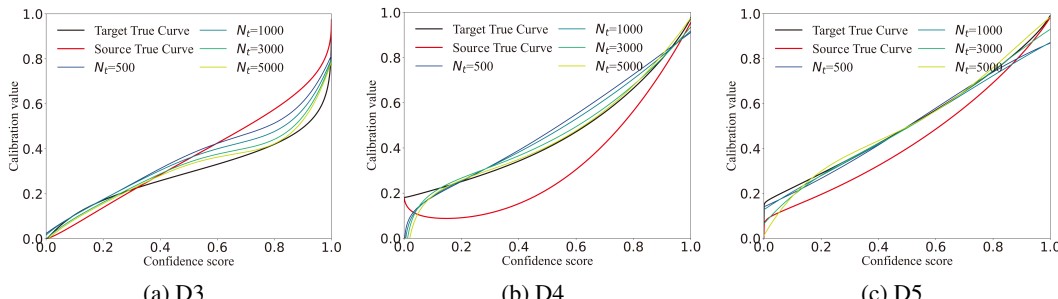

Figure 7: Impact of Source Domain Sample Size.

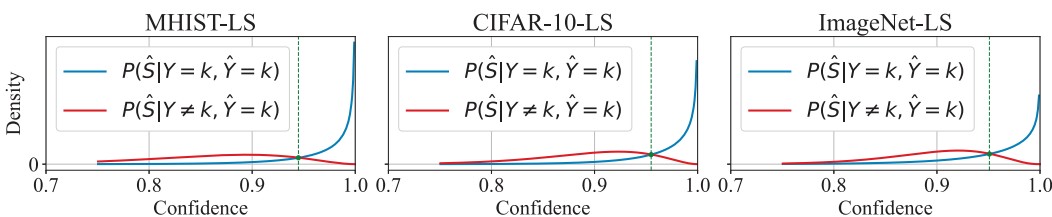

Figure 8: Pole Analysis of Theorem 2. Note that $P(\hat{S} \mid Y = k, \hat{Y} = k) = Q(\hat{S} \mid Y = k, \hat{Y} = k)$ and $P(\hat{S} \mid Y \neq k, \hat{Y} = k) = Q(\hat{S} \mid Y \neq k, \hat{Y} = k)$ due to Eq. 14.

### H.3.5 Impact of Classifier Accuracy on Source Domain

Fig. 9 shows the impact of the classifier's source-domain accuracy on Eq. 7 and TLFCC. As the accuracy of the classifier increases, even if the number of samples with $H = 0$ (i.e., $Y \neq k$ and $\hat{Y} = k$) becomes very small, the calculation of $P(\hat{S} \in b | Y \neq k, \hat{Y} = k)$ will not result in an infinitely large or unstable value. This is because the numerator of Eq. 7 will first become zero compared to the denominator. Furthermore, as shown in Fig. 8(b), the calibration effect of TLFCC will not show any abnormalities when the source-domain accuracy approaches 1.

## I Effectiveness on Out-Domain Scenarios

To test the boundaries of existing confidence calibration methods under label shift, we tested their effectiveness on some out-of-domain datasets where covariate shift and label shift coexist. We selected three datasets commonly used in the covariate shift domain: MNIST (LeCun et al., 2002), USPS (Hull, 1994), and SVHN (Netzer et al., 2011). All three datasets are digit recognition datasets, ensuring class consistency between the source and target domains. In the experiment, two datasets were used as the source domain, and the other dataset was used as the target domain. To simulate label shift, the source domain was sampled as an imbalanced dataset (by adjusting the imbalance factor), and the target domain was sampled as a balanced dataset.

Table 11 compares calibration errors across different methods under out-domain scenarios where both label shift and covariate shift occur. The results show that all methods suffer significant performance degradation compared to in-domain settings, highlighting the difficulty of calibration in such conditions. Among the baselines, LaSCal achieves the best performance but still exhibits high error rates. In contrast, the proposed TLFCC method consistently delivers the lowest calibration error across all target domains (MNIST, USPS, SVHN), network depths, and imbalance factors, reducing ECE by a large margin (e.g., from over 71.80% with LaSCal to about 59.20% on MNIST with IF=10). A potential reason why these methods can improve confidence calibration is that the relationship between covariates and labels remains relatively stable, even in this case, resulting in little change in $P(X|Y)$. These findings confirm that TLFCC remains partially robust and effective even under severe domain shifts without requiring target-domain labels.

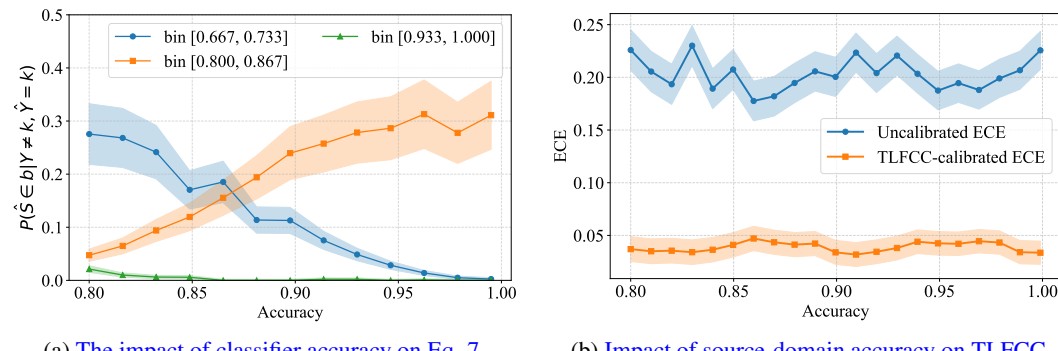

(a) The impact of classifier accuracy on Eq. 7.   (b) Impact of source-domain accuracy on TLFCC

Figure 9: Impact of Source-Domain Accuracy. The experimental data is MHIST-LS, and the classifier is ResNet-18. The reported results are mean ± std over ten runs.

Table 11: Compare calibration errors on out-domain data. "Res" is ResNet (He et al., 2016). The reported results are mean ± std over ten runs.

| Target Domain | $\mathbf{ECE_{bin}}$(%)↓ | | | | | | |
| --- | --- | --- | --- | --- | --- | --- | --- |
| | Uncal | TempScal | PCS | LADE | MRR | LaSCal | TLFCC |
| → *MNIST* | | | | | | | |
| Res20 (IF=2) | $88.40_{\pm2.90}$ | $86.90_{\pm2.40}$ | $89.10_{\pm2.55}$ | $78.20_{\pm1.80}$ | $80.10_{\pm1.95}$ | $74.30_{\pm1.60}$ | $62.80_{\pm0.42}$ |
| Res56 (IF=5) | $86.10_{\pm2.70}$ | $85.70_{\pm2.30}$ | $85.90_{\pm2.40}$ | $76.40_{\pm1.75}$ | $78.90_{\pm1.88}$ | $72.60_{\pm1.55}$ | $60.50_{\pm0.40}$ |
| Res110 (IF=10) | $84.90_{\pm2.60}$ | $84.80_{\pm2.20}$ | $85.30_{\pm2.30}$ | $75.10_{\pm1.70}$ | $77.30_{\pm1.82}$ | $71.80_{\pm1.50}$ | $59.20_{\pm0.38}$ |
| → *USPS* | | | | | | | |
| Res20 (IF=2) | $82.50_{\pm2.80}$ | $83.40_{\pm2.35}$ | $81.70_{\pm2.20}$ | $74.30_{\pm1.65}$ | $75.80_{\pm1.75}$ | $70.40_{\pm1.45}$ | $58.10_{\pm0.36}$ |
| Res56 (IF=5) | $81.10_{\pm2.60}$ | $80.80_{\pm2.20}$ | $80.90_{\pm2.10}$ | $73.10_{\pm1.60}$ | $74.50_{\pm1.68}$ | $69.30_{\pm1.40}$ | $56.70_{\pm0.34}$ |
| Res110 (IF=10) | $80.20_{\pm2.50}$ | $80.60_{\pm2.15}$ | $79.90_{\pm2.05}$ | $72.40_{\pm1.55}$ | $73.80_{\pm1.62}$ | $68.70_{\pm1.38}$ | $55.90_{\pm0.33}$ |
| → *SVHN* | | | | | | | |
| Res20 (IF=2) | $89.30_{\pm3.10}$ | $88.90_{\pm2.60}$ | $89.10_{\pm2.70}$ | $80.50_{\pm1.90}$ | $82.10_{\pm2.00}$ | $76.80_{\pm1.65}$ | $64.20_{\pm0.45}$ |
| Res56 (IF=5) | $87.10_{\pm2.90}$ | $86.70_{\pm2.45}$ | $86.90_{\pm2.50}$ | $78.90_{\pm1.85}$ | $80.30_{\pm1.92}$ | $75.20_{\pm1.60}$ | $62.10_{\pm0.42}$ |
| Res110 (IF=10) | $85.80_{\pm2.80}$ | $85.90_{\pm2.35}$ | $85.10_{\pm2.40}$ | $77.80_{\pm1.80}$ | $79.40_{\pm1.88}$ | $74.10_{\pm1.58}$ | $60.90_{\pm0.41}$ |

## J  DESCRIPTION OF LARGE LANGUAGE MODEL USAGE

We only used the large language model to polish the writing.

