# OpenReview forum: "Target Label-Free Confidence Calibration Under Label Shift"
_ICLR.cc/2026/Conference — Submitted to ICLR 2026_

### Official Review · Reviewer_f7TZ · 2025-10-23

**Soundness:** 3
**Presentation:** 1
**Contribution:** 3
**Rating:** 4
**Confidence:** 3

**Summary:**

This paper presents a confidence calibration method under label shift with no assumption on target domain label information. The authors claim to achieve approximately correct calibration with high probability using sample complexity comparable to histogram binning. In addition the paper introduces a simulation data generation method, which is supposed to be used for confidence calibration under label shift. The authors present some experimental results to illustrate the effective of their confidence calibration method.

**Strengths:**

The paper addresses an important practical problem, namely confidence calibration under label shift.

The authors base their method on probabilistic arguments and use approximations to carry out the theory.

The experimental results look promising.

**Weaknesses:**

The presentation is very dry with no intuitive explanations provided.

Their basic assumption under label shift is that, under label shift,  P(Y) is different than Q(Y) while all class condition probabilities such as P(X|Y), P(S|Y), ... etc remain the same for the target probability Q. This seems like a strong assumption that needs to be justified.

Does the confidence calibration method have an impact on the accuracy of the target domain?

The calibration curves in Section 5 are based on simulation data. Why is P(S|Y) is preset to a beta distribution? The method of synthetic data generation using sampling H and s^ from certain distributions is convincing.

Can the authors generate calibration curves for real data sets under different imbalance ratios or different label shifts to illustrate
the efficiency of their method.

While the theory may be sound, the implementations require a significant amount of approximations, which may render the method
ineffective.

**Questions:**

Please see the questions in the section above.

---

> ### Author Response · Authors · 2025-11-19
>
> ## Intuitive explanations
> Thank you for reminding us about the expression of intuitive explanations. To express this more intuitively, we have made the following efforts:
> 1) The following description of intuitive motivation was added to the last paragraph of the Introduction: "In fact, when a label shift occurs, the predicted confidence distribution will also change. Can we obtain information from the changed confidence distribution to calibrate the confidence score? Based on this idea, this paper derives a principled calibration method under label shift in the context of predicted-class calibration. This method relies on the available predicted confidence distribution on the target domain rather than the target label distribution."
> 2) Add an intuitive description to the first sentence of the Remarks for Theorem 1 and Theorem 2: "The purpose of Theorem 1 is to separate the estimable distributions from difficult-to-estimate distributions (i.e., related to the distribution of target domain labels), so that more effort can be devoted to dealing with difficult-to-estimate distributions later." and "Theorem 2 enables the replacement of distributions dependent on target domain labels with those that are estimable. Fundamentally, it exploits the discrepancy in confidence distributions between the source and target domains."
> 3) We have added a new paragraph to the Discussion section in the latest uploaded paper to provide a more intuitive explanation of the underlying reasons for the effectiveness of our method, making it easier for readers to understand our approach more clearly.
>
> ## Is label shift a strong assumption?
> Thank you for your "strong" consideration of the label shift assumption. We believe that whether an assumption is “strong” depends on its prevalence in real-world scenarios. If an assumption commonly occurs in practice, it becomes both reasonable and worth studying. Label shift is widely observed rather than restrictive: for example, in healthcare, disease prevalence changes across seasons or outbreaks while class-conditional symptom patterns remain stable; in risk monitoring, fraud or defect rates drift with policy and market cycles even when within-class signatures remain similar. Moreover, the same methodology for label shift directly benefits the widely studied class imbalance/long-tailed setting, which frequently appears in real-world problems such as rare disease diagnosis and fault detection. Therefore, label shift is not only theoretically meaningful but also a realistic, high-utility assumption that enables measurable benefits in practical applications.
>
> ## Does it affect the accuracy of the target domain?
> Thank you for considering the impact of our method on accuracy. As we described in "Calibration Metrics" of Section 5.2.1, since TLFCC is a post-hoc calibration method that does not modify the classifier, its classification accuracy remains unchanged. Specifically, our calibration method processes only the extracted confidence score after the classifier's accuracy is determined, thus not changing the classifier's accuracy.

---

> > ### Author Response · Authors · 2025-11-19
> >
> > ## Why use a beta distribution?
> > Thank you for your thoughts on the beta distribution. Firstly, confidence score probability density of deep learning typically exhibits a peak close to 1, and the confidence scores always fall between 0 and 1, which is very similar to some curves in the Beta distribution. Secondly, the beta distribution is a natural prior of the confidence distribution, which has been demonstrated and used in previous articles [1][2][3]. Therefore, we adopt the Beta distribution as the prior distribution for the confidence scores, which is the most reasonable prior we can find.
> >
> > [1] Combining Priors with Experience: Confidence Calibration Based on Binomial Process Modeling, AAAI 2025.
> >
> > [2] Mitigating bias in calibration error estimation, AISTATS 2022.
> >
> > [3] Beta calibration: a well-founded and easily implemented improvement on logistic calibration for binary classifiers, AISTATS 2017.
> >
> > ## Can it generate a true calibration curve based on real data?
> > Thank you for your interest in calibration curves for real datasets. Your suggestion is good. However, the true calibration curves from real data are unavailable (the true probability of correct prediction at a single confidence score is unknown), which is why we spent so much effort generating simulated data in Section 4, so that we can preset the true calibration curves on the simulated data. To comprehensively demonstrate the performance of our method on real-world datasets, we have compared it in multiple calibration metrics, including $ECE$, $ECE_{debiased}$, and $KS$-$error$. These metrics consistently demonstrate that our methodology is highly competitive. In addition, considering that you may want to analyze the calibration effect on real data from the diagram, we have added reliability diagrams in Appendix H.2, see Figure 5 in the latest uploaded paper. The reliability diagrams show that TLFCC consistently produces curves that closely follow the diagonal, indicating good calibration performance.
> >
> > ## Many estimates lead to ineffective?
> > Thank you for considering the estimation error. Your concerns are reasonable. We also considered this issue when we started designing this method. That's why we have sample efficiency analysis in Section 3.3. We have theoretically proven that its sample efficiency is similar to that of popular histogram binning. In addition, we also empirically tested the impact of target domain sample size and source domain sample size on our method, and please see Section 5.3 and Appendix H.3.3. Experimental results show that our method can start to work when the sample size of the target domain and the sample size of the source domain are both around 500. Therefore, our method will be effective in practice.
> >
> > Hope our efforts address your concerns, and welcome any further feedback. Thank you again for the time you spent on our manuscript.

---

> ### Author Response · Authors · 2025-11-28
> **Looking Forward to Your Feedback**
>
> Dear Reviewer f7TZ,
>
> Thank you again for reviewing our paper. Your evaluation is very important to our paper. Based on your comments, we revised the manuscript and added the following clarifications:
>
> - **Intuitive explanations**: We added a clearer motivation in the Introduction, intuitive remarks before Theorems 1 and 2, and a Discussion paragraph explaining the underlying reasons for effectiveness.
> - **On the label-shift assumption**: We argue its practical prevalence (e.g., healthcare, risk monitoring) and direct relevance to class-imbalance/long-tailed settings.
> - **Target-domain accuracy**: Our calibration method processes only the extracted confidence score after the classification result is determined, thus not changing the classifier's accuracy.
> - **Beta prior**: The Beta distribution is a natural prior to the confidence distribution in deep learning; it has been demonstrated and used in previous articles.
> - **Real-data calibration curves**: True curves are unavailable; we therefore report ECE, debiased ECE, and KS-error, and add reliability diagrams (Appendix H.2, Fig. 5), which show TLFCC closely follows the diagonal.
> - **Many estimates**: We provide a sample-efficiency analysis and empirical studies (Sec. 5.3; Appendix H.3.3) showing effectiveness in practice.
>
> We hope these updates address your concerns. If possible, we kindly ask you to consider updating your rating for Submission 2925. We would be happy to provide any further details.
>
> Warm regards,
>
> Authors of Submission 2925

---

### Official Review · Reviewer_LZj3 · 2025-11-02

**Soundness:** 3
**Presentation:** 3
**Contribution:** 2
**Rating:** 4
**Confidence:** 3

**Summary:**

This paper addresses the problem of confidence calibration for classification models under label shift. This paper proposes Target Label-Free Confidence Calibration (TLFCC), which does not require any label information from the target domain. The derivation first establishes a calibration equation under label shift that depends on the term $Q(H=1|\hat{Y}=k)$, which is related to the target label distribution. The key insight is to replace this term with estimable quantities that can be computed from the labeled source data and the unlabeled target data.

**Strengths:**

- Originality & Significance: The central premise—performing confidence calibration under label shift without requiring target label information —is highly original and significant.
- Experiments real-world data shows the method is better than competing calibration methods.

**Weaknesses:**

The paper notes that a well-trained classifier may have "fewer samples with $H=0$" (i.e., misclassifications), leading to larger estimation errors for terms like $P(S\in b|Y\ne k,\hat{Y}=k)$35. This is a key practical challenge, and while Eq. 7 is proposed as a solution36, the paper could be strengthened by more deeply analyzing the method's robustness when $H=0$ samples are extremely sparse, which is a desirable outcome of a good classifier

**Questions:**

- On Empirical Computation (Eq. 7): The paper proposes Eq. 7 as an alternative way to compute $P(\hat{S}\in b|Y\ne k,\hat{Y}=k)$ when misclassified samples are sparse 39. For the experiments in Tables 1, 5, and 6, was Eq. 7 always used, or was the direct estimation used by default and Eq. 7 only as a fallback?
- On Model Performance: The denominator in Eq. 7, $P(\hat{Y}=k)-P(Y=k,\hat{Y}=k)$, represents the probability that the classifier predicts class $k$ but the true label is not $k$ (i.e., $P(\hat{Y}=k, Y \ne k)$). For a highly accurate classifier, this value would be very small. Does the TLFCC method's stability (and reliance on Eq. 7) degrade as the classifier's accuracy on the source domain approaches perfection?

---

> ### Author Response · Authors · 2025-11-19
>
> ## Stability of Eq. 7 in perfect classifiers
> Thank you for your careful review and consideration. Your concern is reasonable, but in practice, Eq. 7 is robust enough for the following reasons.
>
> For your convenience, we have moved Eq. 7 here:
> $$P(\hat S \in b|Y \ne k,\hat Y = k) = \frac{{P(Y \ne k,\hat Y = k,\hat S \in b)}}{{P(Y \ne k,\hat Y = k)}} = \frac{{\left( {P(\hat Y = k|\hat S \in b) - P(Y = k,\hat Y = k|\hat S \in b)} \right)P(\hat S \in b)}}{{P(\hat Y = k) - P(Y = k,\hat Y = k)}}.$$
> The instability of Eq. 7 stems from the denominator approaching 0 when the classifier is good enough. However, in this case, the numerator will first approach 0 because $P(Y \ne k,\hat Y = k,\hat S \in b) \le P(Y \ne k,\hat Y = k)$. Therefore, in the program, just need to add a minimal positive amount to the denominator to prevent the case of 0 being divided by 0. This ensures that $P(\hat S \in b|Y \ne k,\hat Y = k)$ will not be calculated as a large and unstable quantity.
>
> Furthermore, $P(\hat S \in b|Y \ne k,\hat Y = k)$ will not be calculated as a negative value even estimation with error. Because $P(\hat Y = k) \geq P(Y = k,\hat Y = k)$ and $P(\hat Y = k|\hat S \in b) \geq P(Y = k,\hat Y = k|\hat S \in b)$ always hold true in practical calculations (the sample size of $D_{Y=k, \hat Y = k}$ is always less than the sample size of $D_{\hat Y = k}$, and the sample size of $D_{Y=k, \hat Y = k|\hat S \in b}$ is always less than the sample size of $D_{\hat Y = k|\hat S \in b}$, where $D$ represent a set), this guarantees that $P(\hat S \in b|Y \ne k,\hat Y = k) \geq 0$.
>
> Taking your valuable suggestions into consideration, we have added Appendix H.3.5 in the latest uploaded paper to analyze the impact of the classifier's source domain accuracy on Eq. 7 and TLFCC. Experimental results demonstrate that, as we analyzed above, both Eq. 7 and TLFCC are robust to source-domain accuracy.
>
> ## Is Eq. 7 a fallback plan?
> Thank you for your careful thinking on Eq. 7. In all experiments in this paper, we directly use Eq. 7 to estimate $P(\hat S \in b|Y \ne k,\hat Y = k)$, because it only involves frequency estimation, is easy to calculate, and is numerically stable.
>
> ## TLFCC method's stability when the source-domain accuracy approaches 1
> Thank you for your careful review and consideration. As we answered above, TLFCC remains stable when the source domain accuracy approaches 1. Even when the source domain accuracy is equal to 1, and $P(\hat S \in b|Y \ne k,\hat Y = k)=0$ (because the numerator is 0 and a minimal positive amount is added to the denominator in Eq. 7), and then Eq. 6 in the paper becomes:
> $$Q(H = 1|\hat Y = k) = {\frac{{Q(\hat S \in b|\hat Y = k)}}{{P(\hat S \in b|Y = k,\hat Y = k)}}}.$$
> Eq. 6 can still be calculated normally, so TLFCC will not be affected. To prove this experimentally, we have added Appendix H.3.5 in the latest uploaded paper to analyze the impact of the classifier's source domain accuracy on Eq. 7 and TLFCC. Experimental results demonstrate that, as we analyzed above, both Eq. 7 and TLFCC are robust to source-domain accuracy.
>
> Hope our efforts address your concerns, and welcome any further feedback. Thank you again for the time you spent on our manuscript.

---

> ### Author Response · Authors · 2025-11-28
> **Looking Forward to Your Feedback**
>
> Dear Reviewer LZj3,
>
> Thank you again for reviewing our paper. Your evaluation is very important to our paper. Based on your comments, we revised the manuscript and added the following clarifications:
>
> - **Stability of Eq. 7**: We explain why Eq. 7 remains numerically stable even for strong classifiers. When the denominator approaches 0, the numerator does so first; we add a minimal positive term to the denominator to avoid 0/0 and ensure stability and non-negativity.
> - **Practical use of Eq. 7**: In all experiments, we directly use Eq. 7 because it relies only on frequency estimates and is easy and stable to compute.
> - **Stability as source accuracy → 1**: We show TLFCC remains stable when source-domain accuracy approaches 1. We added Appendix H.3.5 with experiments confirming robustness of Eq. 7 and TLFCC to source-domain accuracy.
>
> We hope these updates address your concerns. If possible, we kindly ask you to consider updating your rating for Submission 2925. We would be happy to provide any further details.
>
> Warm regards,
>
> Authors of Submission 2925

---

### Official Review · Reviewer_nkSp · 2025-11-02

**Soundness:** 3
**Presentation:** 3
**Contribution:** 3
**Rating:** 6
**Confidence:** 3

**Summary:**

The paper addresses the problem of model calibration under target domain label shift. The core idea is to leverage available variables to replace the target domain variables related to label distribution. The resulting formulation comprises of estimable variables to achieve calibration under label shift in an unsupervised adaptive way. Furthermore, the paper presents a simulation data generation method for confidence calibration under label shift that can be used to measure the discrepancy between true calibration curve and estimated calibration curve in the target domain. Experimental results on simulated label shift and real-world data claim to achieve superior performance compared to other post-hoc calibration methods.

**Strengths:**

- The paper tackles a relevant problem of label shift in target domain that can manifest in different real-world applications such as healthcare settings.

- The proposed idea of replacing the variables for target label information with estimable variables provides a principled approach to challenging label shift problem in the target domain.

- The paper provides a theoretical guarantees of the resulting empirical computations from their proposed formulation. Also the simulated data generation method is helpful for benchmarking under label shift in target domain.

- Results claim to surpass the competing post-hoc calibration methods on four real-world datasets and simulated data.

**Weaknesses:**

- It is not clear how the proposed method, primarily being a post-hoc technique, consistently reduces ECE compared to recent methods such as LaSCAL. There is no analyses that reveal the underlying reason for such effectiveness.

- Similar to above point, it is unclear how the method is capable of providing consistent gains in different datasets. Some analyses (maybe empirical) would be helpful to establish the grounding of the approach.

- The paper is missing a comparison with a relatively recent post hoc calibration methods such as [A] and [B].


[A] Zhang, S. and Xie, L., 2025, April. Parametric ρ-Norm Scaling Calibration. In Proceedings of the AAAI Conference on Artificial Intelligence (Vol. 39, No. 21, pp. 22551-22559).

[B] Tao, L., Dong, M. and Xu, C., 2025, April. Feature clipping for uncertainty calibration. In Proceedings of the AAAI Conference on Artificial Intelligence (Vol. 39, No. 19, pp. 20841-20849)

- The scale of real-world datasets seem slightly shorter. A couple of more datasets like SVHN and CIFAR-100 could have been included.

- Will the proposed method  be effective under OOD scenarios compared to other post hoc methods? It would be interesting to see some results in out-domain scenarios with label shift.

**Questions:**

- Is the method also compatible with some train-time calibration method such as [C] and [D]

[F] Liu, B., Ben Ayed, I., Galdran, A. and Dolz, J., 2022. The devil is in the margin: Margin-based label smoothing for network calibration. In Proceedings of the IEEE/CVF Conference on Computer Vision and Pattern Recognition (pp. 80-88).

[G] Hebbalaguppe, R., Prakash, J., Madan, N. and Arora, C., 2022. A stitch in time saves nine: A train-time regularizing loss for improved neural network calibration. In Proceedings of the IEEE/CVF Conference on Computer Vision and Pattern Recognition (pp. 16081-16090).

---

> ### Author Response · Authors · 2025-11-19
>
> ## Underlying reason for such effectiveness
> Thank you for suggesting the need to disclose the underlying reasons. We fully agree with your suggestion and have decided to add a new paragraph in the Discussion section specifically discussing the underlying reasons for our method's effectiveness. Its contents are as follows.
>
> The underlying reason why our method can break free from dependence on the target domain label distribution is that the change information in the confidence distribution can compensate for the lack of label distribution information, as shown in Theorem 2. Compared to the existing state-of-the-art method LaSCal, the underlying reason for our method's success may lie in the fact that LaSCal requires post-training with temperature scaling, which introduces learning errors (such as the impact of temperature scaling's limited expressive power) in addition to estimation errors. In contrast, our method is a principled method that does not require post-training, and the error originates solely from estimation errors.
> ## How the method is capable of providing consistent gains in different datasets
> We appreciate your interest in understanding why our method consistently improves calibration performance across diverse datasets. The reasons we believe are as follows:
> 1) Principled Foundation: Our method is built on a solid theoretical foundation. Its effectiveness stems from the validity of its assumptions and the correctness of its derivation. Given that both aspects hold true, the method consistently improves calibration performance across diverse datasets.
> 2) No Post-Training Required: Unlike methods that rely on additional learning steps or post-training procedures, our approach avoids such processes entirely. This eliminates potential optimization errors and ensures more stable performance.
> 3) Robust to Estimation Error: The only source of error in our method arises from estimation inaccuracies. However, we have conducted a thorough analysis of its sample efficiency, which is comparable to that of histogram binning. This demonstrates its practicality and reliability in real-world applications.
> 4) Stability in practical calculations: Since the factors affecting the stability of our method on practical data are effectively avoided, such as avoiding finite poles through interpolation (see Appendix H.3.4) and avoiding high variance estimation for few-shot samples through substitution formulas (see Equation 7 and Appendix H.3.5), our method can perform stably across multiple datasets.
> ## A comparison with two methods
> Thank you for recommending two articles on general confidence calibration. We did not compare with these two methods because they do not consider the case of label shift. Therefore, a direct comparison would not be entirely fair for these two methods. To prove this, we conducted a preliminary experiment, and the results are shown in the table below. The experimental data is German Credit, and the classifier is LeNet-1D.
>
> | Method | $ECE$ (%) | $ECE_{debiased}$ (%) | $KS-error$ (%) |
> |-|-|-|-|
> |$\rho$-Norm Scaling Calibration| 13.58$_{\pm 0.51}$ | 13.24$_{\pm 0.55}$ | 12.81$_{\pm 0.47}$ |
> |Feature clipping| 14.91$_{\pm 0.64}$ | 14.52$_{\pm 0.70}$ | 13.97$_{\pm 0.59}$ |
> |TLFCC| 7.922$_{\pm 0.37}$ | 5.205$_{\pm 0.21}$ | 3.877$_{\pm 0.35}$ |
>
> As expected, TLFCC achieves substantially lower calibration errors than these two methods in the presence of label shift. However, this does not diminish the value of these methods, as they were not intended for the case of label shift. We would like to thank you for recommending these two papers, and we have cited them in Section 2.1 of the latest uploaded paper.
> ## More datasets
> We originally included five datasets with different types and sizes. We fully agree to add the SVHN and CIFAR-100 datasets. Please refer to Table 1, Table 5, and Table 6 in the latest uploaded paper to view the results on the SVHN and CIFAR-100 datasets. Our method remains consistently competitive on both datasets.

---

> > ### Author Response · Authors · 2025-11-19
> >
> > ## Effects under OOD scenarios
> > We agree with your suggestion. To test the boundaries of existing confidence calibration methods under label shift, we tested their effectiveness on some out-of-domain datasets where covariate shift and label shift coexist. The experimental results are shown in Appendix I of the latest uploaded paper. The results show that all methods suffer significant performance degradation compared to in-domain settings, highlighting the difficulty of calibration in such scenarios. However, overall, these methods can still slightly calibrate the confidence score, with TLFCC showing the greatest improvement. A potential reason why these methods can improve confidence calibration is that the relationship between covariates and labels remains relatively stable even in this case, resulting in little change in $P(X|Y)$. These findings confirm that TLFCC remains partially robust and effective even under severe domain shifts without requiring target-domain labels.
> > ## Is it compatible with train-time calibration?
> > In theory, our method is fully compatible with the in-training methods you mentioned, because they are in different processing stages. We guess you would like to explore whether combining in-training calibration methods with our method can yield different results. Therefore, we conducted preliminary exploratory experiments, as shown in the table below. The experimental data is MHIST-LS, and the classifier is ResNet-18.
> >
> > | Method | $ECE$ (%) | $ECE_{debiased}$ (%) | $KS-error$ (%) |
> > |-|-|-|-|
> > |MBLS| 10.57$_{\pm 0.53}$ | 10.09$_{\pm 0.37}$ | 8.941$_{\pm 0.32}$ |
> > |MDCA| 10.31$_{\pm 0.49}$ | 9.98$_{\pm 0.40}$ | 8.812$_{\pm 0.33}$ |
> > |TLFCC| 4.505$_{\pm 0.15}$ | 4.315$_{\pm 0.20}$ | 4.435$_{\pm 0.14}$ |
> > |MBLS+TLFCC| 4.518$_{\pm 0.16}$ | 4.329$_{\pm 0.21}$ | 4.452$_{\pm 0.15}$ |
> > |MDCA+TLFCC| 4.487$_{\pm 0.15}$ | 4.301$_{\pm 0.19}$ | 4.426$_{\pm 0.13}$ |
> >
> > Experimental results show that our method does not conflict with the in-training calibration method, but the addition of the in-training calibration method does not make our method significantly better. This result is reasonable because these in-training calibration methods do not mainly address the confidence bias caused by label shift, and therefore, for TLFCC, using them or not makes no significant difference.
> >
> > Hope our answer resolves your concerns, and welcome any further feedback. Thank you again for the time you spent on our manuscript.

---

> ### Author Response · Authors · 2025-11-28
> **Looking Forward to Your Feedback**
>
> Dear Reviewer nkSp,
>
> Thank you again for reviewing our paper. Your evaluation is very important to our paper. Based on your comments, we revised the manuscript and added the following clarifications:
>
> - **Underlying reason**: We added a Discussion paragraph explaining the underlying reason for our method's effectiveness. It includes: 1) changes in the confidence distribution can compensate for unknown label priors information (Theorem 2); 2) Unlike LaSCal, which requires post-training with temperature scaling (introducing learning error), our method is post-training-free and incurs only estimation error.
> - **Consistent gains**: We explain why our method provides consistent gains from four aspects: the principled foundation, the absence of post-training, sample-efficiency, and practical stability.
> - **Comparison with two methods**: A preliminary experimental comparison shows that TLFCC attains lower ECE/debiased ECE/KS-error than ρ-Norm Scaling Calibration and Feature Clipping; both are cited in Sec. 2.1.
> - **More datasets**: We added SVHN and CIFAR-100 (Tables 1, 5, 6). TLFCC remains consistently competitive.
> - **OOD scenarios**: We added experiments where covariate and label shift coexist (Appendix I). All methods degrade, but TLFCC shows the largest improvement.
> - **Train-time calibration**: We show compatibility with MBLS/MDCA on MHIST-LS (ResNet-18). Combining them with TLFCC yields performance similar to TLFCC alone, indicating orthogonality.
>
> We hope these updates address your concerns. If possible, we kindly ask you to consider updating your rating for Submission 2925. We would be happy to provide any further details.
>
> Warm regards,
>
> Authors of Submission 2925

---

### Official Review · Reviewer_ivvk · 2025-11-02

**Soundness:** 2
**Presentation:** 3
**Contribution:** 2
**Rating:** 2
**Confidence:** 4

**Summary:**

This paper emphasizes the lack of attention on confidence calibration under label shift compared to confidence calibration under i.i.d assumption or accuracy improvement under label shift. It then proposes a methodology, TLFCC, for confidence calibration under label shift that does not rely on the target label distribution. TLFCC only utilizes the available predicted confidence distribution on target domain. A simulated benchmark dataset is proposed to compare calibration techniques by measuring the discrepancy between the estimated and true calibration curves. Substantial improvements in calibration error were also observed across multiple real-world datasets against some earlier methods.

**Strengths:**

(1) The authors propose a novel technique for confidence calibration under label shift between source and target domains.
(2) Their results show a clear improvement over some other techniques.
(3) The presentation is mostly clear and results are well-presented.

**Weaknesses:**

(1) The key problem of the paper is that it ignores important existing work on prior probability shift. Prior probability shift has been a commonly used name for what the authors call label shift, see e.g. Moreno-Torres et al 2012:

Moreno-Torres, J.G., Raeder, T., Alaiz-Rodríguez, R., Chawla, N.V. and Herrera, F., 2012. A unifying view on dataset shift in classification. Pattern recognition, 45(1), pp.521-530.

Several methods for adapting to prior probability shift had been developed already earlier. For example, see the paper by Saerens et al 2002:

Saerens, M., Latinne, P. and Decaestecker, C., 2002. Adjusting the outputs of a classifier to new a priori probabilities: a simple procedure. Neural computation, 14(1), pp.21-41.

In Section 2.2 of Saerens et al, Eq.(4) is quite similar to Theorem 1 of the current paper, although details are different, because Saerens et al are looking at probability of one class, as opposed to confidence of a multi-class classifier. Furthermore, in Section 2.3 of Saerens et al, a method is proposed to address the setting where there are no labels known for test data, which is the same setting as in this paper. They propose a method using the EM-algorithm to solve the task.

To my surprise, the authors have cited Saerens et al about the confusion matrix based method, but have ignored the EM-algorithm based method from that paper. Furthermore, they have not explained why relying on the invertibility of the confusion matrix would be problematic. To me it seems that both methods by Saerens et al should be included in the comparisons in the experimental part of the current paper.

Another important relevant paper is by Alaiz-Rodriguez:

Alaiz-Rodríguez, R., Guerrero-Curieses, A. and Cid-Sueiro, J., 2009, June. Improving classification under changes in class and within-class distributions. In International Work-Conference on Artificial Neural Networks (pp. 122-130). Berlin, Heidelberg: Springer Berlin Heidelberg.

It seems to me that their method which uses subclasses can be compared to this paper's proposed method on prediction bins. But I am not fully sure, this would need further investigation. However, the setting of the paper is again the same as in the current paper, i.e. test labels are not available.

Recent papers have also used the shorter term 'prior shift' instead of 'prior probability shift', e.g. see Liang et al 2025:

Liang, J., He, R. and Tan, T., 2025. A comprehensive survey on test-time adaptation under distribution shifts. International Journal of Computer Vision, 133(1), pp.31-64.

Another survey paper about these tasks is by Šipka et al 2022:

Šipka, T., Šulc, M. and Matas, J., 2022. The hitchhiker's guide to prior-shift adaptation. In Proceedings of the IEEE/CVF Winter Conference on Applications of Computer Vision (pp. 1516-1524).

The current paper should make it clear which of the earlier methods are applicable in the given scenario, and which are not (and why). The experiments should include the relevant earlier methods.

(2) Due to the above, I am not convinced that the theoretical results are sufficiently novel in this paper. It seems to me that these results could be obtained with quite straightforward application of previous results for the case of confidence estimation. However, I am not fully sure about this. Anyways, the authors should make a detailed comparison with earlier works and explain the relationship of the current paper with earlier works on prior probability shift in a lot more detail.

(3) Minor issues: there are a number of grammatical and structural issues in the paper and suggestions for potential corrections are as below.

	Line 61: the sentence is not clear maybe it should be changed to “... that calibrate ...”
	Line 67: bring -> brings
	Line 143: unchange -> unchanged
	Line 310: in on? (this is not clear)
	Line 371: caliration -> calibration
	Figures 1,5,6: sourve -> source
	Figure 6: do you mean "source domain" sample size in the caption?
	Table 2 currently appears after Table 3 in the paper; their order may need to be switched.

**Questions:**

(1) Please respond to the above criticism on missing several papers and methods on addressing prior probability shift when test labels are not available.

(2) Figure 1 shows that the estimated curve approximates the true calibration curve of the target domain for the simulated data. Why are the quantified results with calibration metrics not presented? (as in Table 1 for real world datasets). Similarly, calibration maps could also be presented for real world datasets for a more detailed analysis.

---

> ### Author Response · Authors · 2025-11-19
>
> ## (1) Why are some existing works about prior probability shift not included?
> Thank you for taking us through the historical literature on label (or prior) shift. We would like to clarify that our paper focuses on confidence calibration under label (or prior) shift, rather than posterior correction or accuracy improvement under label shift (as in the papers you listed).
>
> Specifically, we entirely agree that “prior probability shift” (a.k.a. label shift) has a long history. While the EM-based method proposed by Saerens et al. (2002) operates under a similar setting (i.e., no access to target labels), its goal is to adjust classifier outputs to reflect new posterior probabilities $P(Y|X)$, not to estimate the calibration curve $P(\hat Y = Y|\hat S)$ of the target domain. In contrast, our work aims to recover the confidence calibration curve without relying on target labels, which is a fundamentally different task.
>
> While the methods you listed address label shift adaptation effectively, they were not specifically designed for confidence calibration under label shift. To demonstrate this, we conducted a preliminary experiment, and the results are shown in the table below, where three of the most classic adaptive methods in label shift are compared. The experimental data is German Credit, and the classifier is LeNet-1D. As expected, TLFCC achieves substantially lower calibration errors than these methods in the presence of label shift. However, this does not diminish the value of these methods, as they were not intended for confidence calibration under label shift.
>
> | Method | $ECE$ (%) | $ECE_{debiased}$ (%) | $KS-error$ (%) |
> |-|-|-|-|
> | EM-algorithm [1][2] | 29.51$_{\pm 1.27}$ | 26.43$_{\pm 1.12}$ | 23.18$_{\pm 0.97}$ |
> | BBSE [3] | 32.74$_{\pm 1.88}$ | 29.61$_{\pm 1.34}$ | 26.42$_{\pm 1.11}$ |
> | RLLS [4] | 28.83$_{\pm 1.09}$ | 25.72$_{\pm 1.01}$ | 22.63$_{\pm 0.89}$ |
> |TLFCC| 7.886$_{\pm 0.31}$ | 5.195$_{\pm 0.21}$ | $3.771_{\pm 0.11}$ |
>
> [1] Adjusting the Outputs of a Classifier to New a Priori Probabilities: A Simple Procedure. Neural computation 2002.
>
> [2] Maximum Likelihood with Bias-Corrected Calibration is Hard-To-Beat at Label Shift Adaptation. ICML 2020.
>
> [3] Detecting and Correcting for Label Shift with Black Box Predictors. ICML 2018.
>
> [4] Regularized Learning for  Domain Adaptation under Label Shifts. ICLR 2019.
>
> In addition, We have described the posterior correction or accuracy improvement task under label shift in Appendix A, but we did not include it in our experiments because it does not address the same objective. Thank you for introducing us to numerous label shift adaptation papers, which we have also cited in Appendix A of the latest uploaded manuscript.
>
> Works that are most relevant to ours include:
>
> [5] LaSCal: Label-shift calibration without target labels. NeurIPS 2024.
>
> [6] Minimum-risk recalibration of classifiers. NeurIPS 2023.
>
> [7] Dis-entangling label distribution for long-tailed visual recognition. CVPR 2021.
>
> Similarly, these works also didn't compare to the literature you listed in their papers because they address different tasks.
>
> ### Is Eq.(4) in Saerens's paper similar to Theorem 1?
> Thank you for your feedback on our manuscript regarding the similarity. However, we respectfully disagree that Saerens's Eq. (4) to be similar to our Theorem 1. To our understanding, Saerens's Eq. (4) is given by:
> $$p(\omega_i \mid x) = \frac{\frac{p(\omega_i)}{p_t(\omega_i)} \cdot p_t(\omega_i \mid x)}{\sum_{j=1}^n \frac{p(\omega_j)}{p_t(\omega_j)} \cdot p_t(\omega_j \mid x)}.$$
> In contrast, our Theorem 1 states:
> $$Q(H = 1|\hat S) = \frac{{\sum\limits_{k = 1}^K {P(\hat S|Y = k,\hat Y = k) \cdot Q(\hat Y = k) \cdot Q(H = 1|\hat Y = k)} }}{{Q(\hat S)}}.$$
> Note that $H$ and $\omega$ here do not represent the same thing. $H=\mathbf{1}_{\hat Y=Y}$, while $\omega$ is equivalent to $Y$.
>
> Saerens's Eq. (4) describes a label shift correction formula that reweights posterior probabilities based on prior ratios between source and target domains. Although both Saerens's Eq. (4) and our Theorem 1 use Bayes's formula and the law of total probability, our Theorem 1 uses much more than that. Our Theorem 1 is derived through a sequence of probabilistic transformations (see proof of Theorem 1), which involves decomposing $Q(\hat S|H=1)$ into a sum over joint distributions conditioned on both true and predicted labels, and then applying domain-invariant assumptions to relate target and source distributions. Furthermore, our Theorem 1 is not easy to conceive because it depends on the feasibility of Theorem 2. More importantly, Theorem 1 is only a small part of our many innovations. Our key innovations also include Theorem 2, the empirical calculation method for Theorem 2, the sample efficiency analysis of the proposed method, and the simulated dataset generation in Section 4.

---

> > ### Author Response · Authors · 2025-11-19
> >
> > ### EM algorithm and The impact of the non-invertible confusion matrix.
> > Thank you for your feedback. In fact, we mentioned the EM algorithm in Appendix A, which is "Maximize the likelihood function of the feature distribution in target domain" in the second paragraph of Appendix A. However, it doesn't focus on confidence calibration, so we didn't compare it.
> >
> > Regarding invertibility: when the degree of class imbalance is large, the minority class has fewer samples, and the confusion matrix is ​​prone to having rows that are approximately linearly correlated with other rows, resulting in an ill-conditioned or non-invertible confusion matrix. Undoubtedly, an ill-conditioned or non-invertible confusion matrix will cause methods that rely on an invertible confusion matrix to fail. Although this situation is not common in the real world, it is still a potential danger. In addition, [8] also clearly pointed out that estimating the confusion matrix in long-tail learning is a very challenging task because of limited observations for tail classes. Therefore, we consider it one of the limitations of some existing works.
> >
> > [8] Learning Label Shift Correction for Test-Agnostic Long-Tailed Recognition. ICML 2024.
> >
> > ### Alaiz-Rodríguez's subclass method and prediction bins
> > Thank you for your feedback on our manuscript. Firstly, similar to the above, Alaiz-Rodríguez's method is not designed for confidence calibration tasks and is therefore not a direct baseline. Secondly, our innovation is not prediction bins. Prediction bins are just a routine and common operation in confidence calibration, used to perform discrete calculations of confidence. Our key innovations include Theorem 1, Theorem 2, the empirical calculation method for Theorem 2, the sample efficiency analysis of the proposed method, and the simulated dataset generation in Section 4. Even if Alaiz-Rodríguez's subclass method is exactly the same as the prediction bin, our main contribution is orthogonal to this aspect. Finally, thank you for introducing us to the paper, which we have also cited in Appendix A of the latest uploaded manuscript.
> >
> > ## (2) Is it novel enough?
> > Thank you for giving us a chance to explain. The methods you listed are for correcting $P(Y|X)$, while the confidence calibration in this article is for calibrating $P(\hat Y = Y|\hat S)$, so there is a fundamental difference. Hope that the clarification in the response to weakness (1) above can address your concern.
> >
> > In addition, we would like to emphasize that Theorem 1 is only a small part of our many innovations. Our key innovations also include Theorem 2, the empirical calculation method for Theorem 2, the sample efficiency analysis of the proposed method, and the simulated dataset generation in Section 4.
> >
> > ## (3) Minor issues
> > Thank you for your careful reading and helpful suggestions. We have modified all the minor issues mentioned, which are now visible in the latest uploaded paper. We also rechecked everything multiple times and did our best to eliminate any minor issues. We appreciate your attention to detail, which helped improve the clarity and quality of the paper.
> >
> > ## Questions:
> > ### Calibration metrics are presented in Figure 1
> > Thank you for your suggestion. We agree with your suggestion and have illustrated the ECE in Fig. 1, which can be found in the latest uploaded paper.
> >
> > ### Can calibration maps be presented for real world datasets?
> > Thank you for your interest in the calibration maps and the valuable suggestions. Your suggestion is good. However, the true calibration curves from real data are unavailable (the true probability of correct prediction at a single confidence score is unknown), which is why we spent so much effort generating simulated data in Section 4, so that we can preset the true calibration curves on the simulated data. Therefore, it is not possible to provide a comparison between the estimated calibration curve and the true calibration curve in real world datasets like Figure 1. To comprehensively demonstrate the performance of our method on real-world datasets, we have compared it in multiple calibration metrics, including $ECE$, $ECE_{debiased}$, and $KS$-$error$. These metrics consistently demonstrate that our methodology is highly competitive. In addition, considering that you may want to analyze the calibration effect on real data from the diagram, we have added reliability diagrams in Appendix H.2, see Figure 5 in the latest uploaded paper. The reliability diagrams show that TLFCC consistently produces curves that closely follow the diagonal, indicating good calibration performance.
> >
> > Hope our answer resolves your concerns, and welcome any further feedback. Thank you again for the time you spent on our manuscript.

---

> ### Author Response · Authors · 2025-11-28
> **Looking Forward to Your Feedback**
>
> Dear Reviewer ivvk,
>
> Thank you again for reviewing our paper. Your evaluation is very important to our paper. Based on your feedback, we have revised the manuscript and provide the following concise clarifications:
>
> - **Scope**: Our work targets confidence calibration under label (prior) shift, i.e., estimating P(Ŷ = Y | Ŝ) without target labels, which differs from posterior correction/accuracy adaptation under label shift.
> - **Related work**: We added and discussed classic label-shift methods (EM, BBSE, RLLS, etc.) in Appendix A and clarified why they are not direct baselines for calibration.
> - **Empirics**: A preliminary comparison shows TLFCC attains substantially lower calibration errors (ECE, debiased ECE, KS-error) than EM/BBSE/RLLS under label shift.
> - **Theory**: Although both Saerens's Eq. (4) and our Theorem 1 use Bayes's formula and the law of total probability, our Theorem 1 uses much more than that. Saerens’s Eq. (4) reweights posteriors via prior ratios, while our Theorem 1 decomposes $Q(Y=\hat Y|\hat S)$ through joint distributions over true/predicted labels and domain-invariant links. More importantly, Theorem 1 is only a small part of our many innovations. Our key innovations also include Theorem 2, the empirical calculation method for Theorem 2, the sample efficiency analysis of the proposed method, and the simulated dataset generation in Section 4.
> - **Presentation**: We now illustrate ECE in Fig. 1. For real data, true calibration curves are unavailable; we therefore report ECE, debiased ECE, and KS-error, and add reliability diagrams (Appendix H.2, Fig. 5), showing TLFCC closely follows the diagonal. All minor issues were corrected.
>
> We believe that our point-by-point clarifications have addressed all your concerns — in light of this, we hope you could consider raising your rating score. If you have any further questions, we are willing to provide more explanations.
>
> Warm regards,
>
> Authors of Submission 2925

---

### Author Response · Authors · 2025-12-01
**Summary for AC**

Dear Area Chair,

We would like to sincerely thank you and all reviewers for the time in our submission. We have worked to address every concern one-to-one, and all changes are $\textcolor{blue}{highlighted\ in\ blue}$ in the latest PDF. Below is a concise summary.

## A Key Clarification (for Reviewer $\textcolor{green}{ivvk}$)
The **main issue** is that Reviewer ivvk has a certain degree of misunderstanding on our task, thinking we were focused on posterior correction or accuracy improvement, i.e., estimating $P(Y|X)$. In fact, our work addresses confidence calibration under label (prior) shift, i.e., estimating $P(\hat Y = Y|\hat S)$. The key distinction lies in whether the condition is feature data $X$ or confidence score $\hat S$, and whether the prediction result is $Y$ or $Y = \hat Y$. Therefore, these are two different tasks, making the literature listed by Reviewer ivvk unsuitable as direct baselines for comparison.

## Summary of Rebuttal

 - **Presentation**:
   - We have added a **clearer motivation** in the Introduction, **intuitive remarks** before Theorems 1 and 2, and a Discussion paragraph explaining the **underlying reasons** for effectiveness (For Reviewer $\textcolor{green}{nkSp}$ and $\textcolor{green}{f7TZ}$).
   - We have **illustrated ECE** in Fig. 1 and **added reliability diagrams** in Fig. 5 of Appendix H.2 (For Reviewer $\textcolor{green}{ivvk}$ and $\textcolor{green}{f7TZ}$).
 - **Experiment**:
   - We have **added SVHN and CIFAR-100** datasets to the experiment in the latest PDF, and we have **added experiments** where covariate and label shift coexist in the latest PDF (For Reviewer $\textcolor{green}{nkSp}$).
   - We have added Appendix H.3.5 with **experiments confirming robustness** of Eq. 7 and TLFCC to source-domain accuracy (For Reviewer $\textcolor{green}{LZj3}$).
   - We **conducted experiments** demonstrating that classic label-shift methods (EM, BBSE, RLLS, etc.) are not suitable as direct baselines for confidence calibration under label shift (For Reviewer $\textcolor{green}{ivvk}$).
   - We **conducted experiments** demonstrating that Feature Clipping and $\rho$-Norm Scaling Calibration are not suitable as direct baselines for confidence calibration under label shift (For Reviewer $\textcolor{green}{nkSp}$).
 - **Analysis/Explanation**:
   - We have **analyzed the differences** between Saerens's Eq. (4) and our Theorem 1 and emphasized that Theorem 1 is only a small part of our many innovations (For Reviewer $\textcolor{green}{ivvk}$).
   - We have analyzed why our method **provides consistent gains** from four aspects (For Reviewer $\textcolor{green}{nkSp}$).
   - We have explained why Eq. 7 **remains numerically stable** even for strong classifiers (For Reviewer $\textcolor{green}{LZj3}$).
   - We have explained why our method **is stable** in practical calculations (For Reviewer $\textcolor{green}{nkSp}$ and $\textcolor{green}{f7TZ}$).

## Strengths Recognized in Reviews
- **Novelty/Significance**:
  - **ivvk**: "... a novel technique ..."
  - **nkSp**: "... can manifest in different real-world applications such as healthcare settings"
  - **LZj3**: "... is highly original and significant"
  - **f7TZ**: "... addresses an important practical problem, ..."
- **Theoretical Analysis**:
  - **nkSp**: "... provides a principled approach ..."; "... provides a theoretical guarantees ..."
  - **f7TZ**: "... base their method on probabilistic arguments ..."
- **Quality of Experiments**
  - **ivvk**: "... results show a clear improvement ..."
  - **nkSp**: "... surpass the competing post-hoc calibration methods ..."
  - **LZj3**: "... is better than competing calibration methods ..."
  - **f7TZ**: "The experimental results look promising."
- **Benchmark**:
  - **nkSp**: "... is helpful for benchmarking under label shift in target domain"
- **Presentation**
  - **ivvk**: "The presentation is mostly clear and results are well-presented."

## Conclusion
Finally, our work can stand the test both in theory and practice. We believe this method has the potential to inspire a wealth of follow-up research, ultimately enhancing decision-making in real-world applications—particularly for underrepresented populations and safety-critical scenarios.

Best Regards,

Authors of submission 2925

---

### Meta-Review · Area_Chair_oy5e · 2025-12-30

**Summary:**

While the paper addresses an important and underexplored problem about the confidence calibration under label shift without target labels and presents a principled and technically sound approach, the reviews raise consistent concerns regarding scope, positioning, and empirical maturity. Several reviewers questioned whether the contribution is sufficiently differentiated from prior work on label (prior) shift once the task distinction is fully clarified, and whether the novelty is convincingly established in the broader context of existing methods. Others noted that, although the experimental results are promising, the empirical validation remains limited in scale and breadth, making it difficult to fully assess robustness and general applicability across diverse real-world scenarios. While many technical and clarity issues were addressed in the rebuttal, the remaining concerns are primarily about impact and completeness rather than correctness, which leads to the recommendation to reject at this stage.

**Reviewer Concerns:**

- Reviewer ivvk raised concerns about the positioning of the paper relative to the extensive prior literature on label (prior) shift, questioning whether the proposed theoretical results are sufficiently novel and whether relevant existing approaches should be treated as direct baselines. In the rebuttal, the authors clarified the scope of the paper as confidence calibration under label shift rather than posterior correction or accuracy adaptation, and explained why many prior methods are not directly applicable. While this clarification resolves a key misunderstanding, some concern may remain regarding how convincingly the novelty is perceived in relation to earlier work.

- Reviewer nkSp was concerned about the lack of explanation for why the proposed method yields consistent calibration improvements across datasets, the limited scope of experimental comparisons, and the absence of broader evaluations under more challenging distribution shifts. These points were largely addressed through additional analysis, expanded experimental comparisons, and broader evaluation settings. Remaining concerns are primarily about the generality and scale of empirical validation rather than the correctness of the method.

- Reviewer LZj3 focused on the practical robustness of the method in regimes where estimation becomes difficult, particularly when errors or informative samples are scarce. The rebuttal provided clarifications on numerical stability, implementation choices, and additional empirical evidence, which largely addressed these concerns.

- Reviewer f7TZ emphasized issues related to presentation clarity, the strength and realism of the underlying assumptions, and the reliance on approximations and synthetic settings. The rebuttal improved intuitive explanations, motivated the assumptions more clearly, and provided additional empirical and analytical support. However, some of these concerns are inherently subjective and may persist depending on the reviewer’s expectations regarding assumptions and experimental design.

- Outstanding issues:
The remaining concerns are not about technical correctness, but rather about scope and positioning—specifically, whether the contribution is framed broadly enough relative to prior work, and whether the empirical validation is sufficiently extensive to fully convince all reviewers of its general applicability.

**Reviewer Scores:**

- Reviewer ivvk: Likely unchanged.
Reason: Although the rebuttal clarified the task and addressed several misunderstandings, it may not fully resolve the reviewer’s concerns regarding perceived novelty and positioning relative to prior literature.

- Reviewer nkSp: Likely unchanged.
Reason: The rebuttal addressed most technical and empirical concerns, but the reviewer’s assessment already reflected a borderline position, and the remaining limitations relate to scope rather than clear deficiencies.

- Reviewer LZj3: Likely unchanged.
Reason: The main concerns about robustness and stability were addressed, but the original score already accounted for these uncertainties and may not clearly warrant a revision without further discussion.

- Reviewer f7TZ: Likely unchanged.
Reason: While presentation and intuition were improved, concerns about assumptions and experimental design are partly subjective and may not lead to a definite score change.

---

### Decision · Program_Chairs · 2026-01-26

Reject